

# Identification of soil-cooling rains in southern France from soil temperature and soil moisture observations

Sibo Zhang[1, 2], Catherine Meurey[1], and Jean-Christophe Calvet[1]

[1]CNRM (Université de Toulouse, Meteo-France, CNRS), Toulouse, France

[2]now at: Qian Xuesen Laboratory of Space Technology, China Academy of Space Technology (CAST), Beijing, China

*Correspondence to*: Jean-Christophe Calvet (jean-christophe.calvet@meteo.fr)

**Abstract.** In this study, the frequency and intensity of soil-cooling rains is assessed using in situ observations of atmospheric and soil profile variables in southern France. Rainfall, soil temperature and topsoil volumetric soil moisture (VSM) observations, measured every 12 minutes at 21 stations of the SMOSMANIA (Soil Moisture Observing System -

Meteorological Automatic Network Integrated Application) network, are analyzed over a time period of 9 years, from 2008 to 2016. The spatial and temporal statistical distribution of the observed rainfall events presenting a marked soil-cooling effect is investigated. It is observed that the soil temperature at a depth of 5 cm can decrease by as much as 6.5 ℃ in only 12 minutes during a soil-cooling rain. We define marked soil-cooling rains as rainfall events triggering a drop in soil temperature at a depth of 5 cm larger than 1.5 °C in 12 minutes. Under Mediterranean and Mediterranean-mountain climates,

it is shown that such events occur up to nearly 3 times a year, and about once a year on average. This frequency decreases to about once every 3.5 years under semi-oceanic climate. Under oceanic climate, such pronounced soil-cooling rains are not observed over the considered period of time. Rainwater temperature is estimated for 13 cases of marked soil-cooling rains using observed changes within 12 min in soil temperature at a depth of 5 cm, together with soil thermal properties and changes in VSM. On average, the estimated rainwater temperature is generally lower than the observed ambient air

temperature, wet-bulb temperature, and topsoil temperature at a depth of 5 cm, with mean differences of -5.1 ℃, -3.8 ℃, and -11.1 ℃, respectively. The most pronounced differences are attributed to hailstorms or to hailstones melting before getting to the soil surface. Ignoring this cooling effect can introduce biases in land surface energy budget simulations.

## 1 Introduction

Over natural and agricultural land surfaces, the frequency and intensity of rainfalls govern soil moisture dynamics from

topsoil layers to the root-zone. While these processes are represented in land surface models (LSMs) (e.g. Decharme et al., 2013), sensible heat input from liquid water into the soil and its impact on the soil temperature profile is often overlooked (Wang et al., 2016). Rainwater temperature is rarely measured and raindrop temperature is not explicitly simulated in atmospheric models (Wei et al., 2014).

The impact of neglecting this process was investigated in recent studies, in the context of global atmospheric simulations.

Wang et al. (2016) showed that the impact on land surface temperature was relatively small in LMDZ simulations (less than



0.3 °C) but they considered mean annual air temperature differences only. Wei et al. (2014) showed that accounting for precipitation-induced sensible heat helps reducing biases in land surface air temperature simulated by the Community Earth System Model 1 (CESM1). Focusing on the winter season in the northern hemisphere, they found a pronounced effect (up to 1 °C) on land surface air temperature in their simulations. In both studies, it was assumed that rainwater temperature was

5 equal to the air wet-bulb temperature. This assumption is valid for rain drops in thermal equilibrium with the ambient air (Kinzer and Gunn, 1951). Both studies tended to show a soil-cooling effect over France but for some regions at higher latitudes a soil-warming effect was simulated.

At the field scale, Kollet et al. (2009) simulated the impact of a rainfall event on soil temperature at a depth of 5 cm ($T_{5cm}$) over a grassland site in the Netherlands. They showed that accounting for precipitation-induced sensible heat in their

simulations triggered a drop in $T_{5cm}$ of more than 1 °C on a single rainfall event. The simulated surface energy fluxes were markedly influenced by this soil-cooling event during several days. In particular, the ground heat flux was affected during nearly one month. They suggested that this effect could be more pronounced in regions affected by strong convective rainfall events. They did not measure rainwater temperature and they assumed that it was equal to the air wet-bulb temperature.

Rainwater temperature was measured in 1947 by Byers et al. (1949) at Wilmington, Ohio, USA. They analyzed the data

from 7 storms and concluded that 3 types of events could be distinguished according to rainwater temperature values within minutes after the rain onset: close to air temperature, close to wet-bulb temperature, and lower than air temperature by as much as 10 °C. They attributed the latter kind of event to hail melting before reaching the surface.

More than 70 years after Byers et al. (1949) performed their experiment, we were not able to find other examples in the open literature of the analysis of ground rainwater temperature observations. Such measurements are usually not performed in

networks of ground meteorological stations.

The objectives of this study are to assess (1) the frequency of soil-cooling rains across contrasting climatic areas in southern France using in situ observations of atmospheric and soil profile variables, (2) the feasibility of estimating rainwater temperature using soil temperature profile measurements, (3) the difference between rainwater temperature and the ambient air temperature or wet-bulb temperature and the topsoil temperature, (4) the impact of neglecting precipitation-induced

sensible heat in the soil temperature simulations of a LSM.

We use the in situ soil temperature and volumetric soil moisture (VSM) measurements collected by the SMOSMANIA (Soil Moisture Observing System - Meteorological Automatic Network Integrated Application) network in southern France (Calvet et al., 2007, 2016), over a 9-year time period from 2008 to 2016. The soil profile measurements are made at a high sampling frequency of one observation every 12 min and the accumulated precipitation is available at the same frequency.

This permits investigating intense precipitation events and their impact on topsoil variables, together with the retrieval of rainwater temperature in certain conditions. We perform and use soil temperature and soil moisture simulations of the ISBA (Interactions between Soil, Biosphere, and Atmosphere) LSM to show the impact on the simulated topsoil temperature of neglecting the precipitation-induced sensible heat flux.



The observations and the model simulations are presented in Sect. 2. Methods for selecting marked soil-cooling rains and for estimating rainwater temperature are described in Sect. 3. The statistical distribution of soil-cooling rains, and estimates of rainwater temperature are presented in Sect. 4. The results are discussed in Sect. 5. The main conclusions are summarized together with prospects for further research in Sect. 6.

## 2 Data

The SMOSMANIA network was implemented in southern France by Meteo-France, the French national meteorological service, in order to monitor in situ soil moisture and soil temperature in contrasting soil and climatic conditions at operational automatic weather stations (Calvet et al., 2007). The SMOSMANIA network is composed of 21 stations forming an Atlantic-Mediterranean transect shown in Fig. 1. Soil and climate types for the 21 stations of the SMOSMANIA network are presented in Table 1. Station full names are given in Table S1 (see Supplement). The 3 westernmost stations are close to the Atlantic Ocean and present an oceanic climate. Further east, 6 stations present a semi-oceanic climate. The 12 easternmost stations are close to the Mediterranean sea and present a Mediterranean climate. Among them, five are located at altitudes above sea-level (a.s.l.) higher than 400 m a.s.l. over complex, mountainous terrain, in the Corbières (MTM) and Cévennes (LGC, MZN, BRN, BRZ) mountainous areas. Two stations (MZN and BRZ) of the Cévennes area are located higher than 600 m a.s.l. and present lower mean monthly minimum and maximum temperatures ($T_{min}$ and $T_{max}$, respectively) than the other stations. While $T_{min}$ can be 2 to 3 °C below freezing level at wintertime at MZN and BRZ, $T_{min}$ is always higher than 1 to 5 °C above freezing level at the other stations. The $T_{max}$ values do not exceed 21 °C and 24 °C at MZN and BRZ, respectively, while $T_{max}$ at the other stations can reach 26 to 30 °C. Oceanic, semi-oceanic, and Mediterranean climates are characterized by contrasting precipitation regimes, with maximum monthly precipitation occurring at wintertime, at spring, and in the autumn, respectively (see Sect. 4.1). It must be noticed that Mediterranean stations are often affected by severe convective precipitation events such as thunderstorms and hailstorms, especially in the autumn (Ruti et al., 2016). This is true for stations located in mountainous areas, but also for those located in the foothills of Corbières (NBN), Cévennes (PZN, PRD, VLV, MJN), and Monts de Vaucluse (CBR).

In general, the soil around the stations is covered by grass. The soil properties were measured for each station as described in Calvet et al. (2016). In the SMOSMANIA network, VSM and soil temperature are measured every 12 min at four depths (5, 10, 20 and 30 cm) using ThetaProbe and PT100 sensors, respectively. The soil moisture (temperature) observations are recorded with a resolution of 0.001 m³ m⁻³ (0.1 °C). The data are available to the research community through the International Soil Moisture Network website (ISMN, 2018). In this study, the sub-hourly observations of VSM and soil temperature were used over 9 years from 2008 to 2016.

Additionally, the SMOSMANIA network consists of preexisting automatic weather stations operated by Météo-France, measuring atmospheric variables. We used a number of meteorological observations such as the maximum and minimum air temperatures in an hour at 2 m, the hourly mean relative humidity (RH) of the air, and the accumulated rainfall every 12 min. A small fraction (less than 4 %) of the rainfall data is missing at each station. For most of the stations, a larger fraction of the



VSM and soil temperature observations is missing. The mean fractions of missing data for all stations are 0.11 and 0.15 for VSM and $T_{5cm}$, respectively. The mean fraction of missing data for either VSM or $T_{5cm}$ is 0.17. More details on missing data for each station, including the seasonal distribution of missing data, are given in Figs. S1, S2, and in Table S1 (see the Supplement). For stations presenting a fraction of missing data larger than 0.1, the fraction of missing data is relatively

evenly distributed across seasons. Missing data are slightly more frequent at wintertime and at spring. The maximum fraction of missing VSM at 5 cm is 0.23 at the SBR station, and the maximum fraction of missing $T_{5cm}$ is 0.47 at the VLV station (Fig. S1). The scaled missing data fraction of either VSM or $T_{5cm}$ is shown in Fig. S2 and Table S1 for each season.

In addition to in situ observations, soil temperature and VSM simulations at depths of 5, 10, 20 and 30 cm were performed using the ISBA (Interactions between Soil, Biosphere, and Atmosphere) LSM within the SURFEX (version 8.0) modeling

platform (Masson et al., 2013). The ISBA configuration and the hourly SAFRAN atmospheric analysis (Durand et al., 1993, 1999) we used to force the model at a spatial resolution of 8 km x 8 km are described in Lafont et al. (2012) and Decharme et al. (2013). In this study, ISBA simulation of topsoil soil moisture and soil temperature profiles for the grassland plant functional type are considered.

## 3 Methods

### 3.1 Identification of intense soil-cooling

In situ rainfall observations are used to identify various types of rainfall events in six steps summarized in Table 2. In a first stage, a rainfall event is defined as a continuous time series of non-zero accumulated liquid precipitation values at time intervals of 12 minutes. Then we only keep the fully documented rainfall events with available soil temperature and VSM observations at a depth of 5 cm. Another stage of data sorting (step 3 in Table 2) is needed to ensure that rainwater is not

completely intercepted by vegetation and is able to actually reach the soil. Because of the method used to count rainfall events, only about 5 % of the rainfall events exceed 5 mm and have a significant impact on the topsoil VSM. Among these marked rainfall events, we sort out (step 4) those affecting $T_{5cm}$ by more than 1 °C. In order to assess the precipitation-induced sensible heat flux on the topsoil temperature, we then select marked soil-cooling rainfall events (step 5) presenting at least one marked drop of $T_{5cm}$ of at least -1.5 °C in 12 min. No other meteorological factor can trigger such rapid changes

in topsoil temperature.

Finally, a few cases are selected (step 6) for the assessment of the rainwater temperature retrieval (Sect. 3.2), in conditions where the topsoil VSM profile is sufficiently affected by the rainwater.

Since a noticeable fraction of observed $T_{5cm}$ or $VSM_{5cm}$ data is missing, the number of marked soil-cooling rainfall events could be underestimated. The missing data fraction across seasons (Table S1) is used to correct the estimation of the possible

number of intense soil-cooling rainfall events, their frequency and the mean time lag between two events.




### 3.2 Estimation of rainwater temperature

In the SMOSMANIA network, rainwater temperature is not measured. We investigate whether it is realistic to postulate that the rainwater temperature can be estimated using in situ topsoil $VSM_{5cm}$ and $T_{5cm}$ observations within a topsoil layer of depth $\Delta z = 0.1$ m. We focus on marked soil-cooling rainfall events corresponding to a drop in topsoil temperature associated to a

rise in VSM values. This condition is not always satisfied in practice because the soil temperature probes and the soil moisture probes at a given soil depth are not located at exactly the same place. In order to limit the impact of spatial heterogeneities in the infiltration of rainwater into the soil, we consider intense precipitation events able to markedly wet the topsoil soil moisture ($VSM_{5cm}$) together with the deeper soil layer ($VSM_{10cm}$) during the drop in $T_{5cm}$. The in situ VSM observations at a depth of 10 cm ($VSM_{10cm}$) are used to ensure that the rainwater really penetrates into the soil and affects the

topsoil layer as a whole. Among 122 rainfall events presenting intense soil-cooling, 101 events have available $VSM_{10cm}$ observations, and only 50 events present an increase in $VSM_{10cm}$ larger than 0.05 $m^3$ $m^{-3}$. Within these 50 rainfall events, we only select 13 events for which the rise in $VSM_{5cm}$ corresponds to the rise in $VSM_{10cm}$ and to the drop in $T_{5cm}$ within the same 12-minute slot (step 6 in Table 2). Firstly, the soil heat capacity at a depth of 5 cm at time $t$ ($C_{5cm}^t$, in units of J $m^{-3}$ $K^{-1}$) is estimated using the volumetric soil moisture at a depth of 5 cm at time $t$, $VSM_{5cm}^t$ (in units of $m^3$ $m^{-3}$):

$$C_{5cm}^t = C_{water} VSM_{5cm}^t + C_{min} f_{min} + C_{SOM} f_{SOM} \qquad (1)$$

where $C_{water}$, $C_{min}$, $C_{SOM}$ are heat capacity values of water, soil minerals, and soil organic matter (SOM) ($4.2\times10^6$, $2.0\times10^6$, and $2.5\times10^6$ J $m^{-3}$ $K^{-1}$, respectively). Their corresponding volumetric fractions at a depth of 5 cm (Table 3) are $VSM_{5cm}^t$, $f_{min}$

and $f_{SOM}$. Soil minerals consist of sand, clay, silt and gravels. Values of the VSM at saturation at a depth of 5 cm $VSM_{sat}$ (the porosity) is also listed in Table 3 for all stations. While the volumetric fractions of sand ($f_{sand}$), clay ($f_{clay}$) and silt ($f_{silt}$) were directly measured at a depth of 5 cm, the volumetric fraction of gravels ($f_{gravel}$) was derived from measurements made at a depth of 10 cm (Sect. 5.1).

During a short time period from time $t_1$ to $t_2$ (12 min in this study) of intense precipitation for which the precipitation-
induced sensible heat flux dominates heat exchanges in the topsoil layer, we can assume that the heat storage change in the topsoil layer, in units of J $m^{-2}$, corresponds to the change in temperature of the infiltrated rainwater as:

$$C_{5cm}^{t_1} \left(T_{5cm}^{t_1} - T_{5cm}^{t_2}\right) \Delta z = C_{water} \left(VSM_{5cm}^{t_2} - VSM_{5cm}^{t_1}\right) \left(T_{rain}^{t_2} - T_{rain}^{t_1}\right) \Delta z \qquad (2)$$

One may assume that soil and incoming rainwater have reached thermal equilibrium at time $t_2$. Houpeurt et al. (1965) showed that thermal equilibrium is nearly instantaneous for small soil mineral particles (e.g. about 10 seconds or less for particle size of less than 5 mm). With this assumption, the rainwater temperature at time $t_2$ ($T_{rain}^{t_2}$) is equal to the measured



soil temperature at time $t_2$ ($T_{5cm}^{t_2}$), and the rainwater temperature just before reaching the soil at time $t_1$ ($T_{rain}^{t_1}$) can be estimated as:

$$T_{rain}^{t_1} = T_{5cm}^{t_2} - \frac{C_{5cm}^{t_1}}{C_{water}} \frac{\left(T_{5cm}^{t_1} - T_{5cm}^{t_2}\right)}{\left(VSM_{5cm}^{t_2} - VSM_{5cm}^{t_1}\right)} \tag{3}$$

It must be noticed that Eqs. (2) and (3) are valid for liquid rainwater only. During hailstorms, hailstones can melt at the soil surface and part of the heat extracted from the topsoil layer (left term in Eq. (2)) is used for ice melting. Hailstones may melt after getting to the soil surface or just before. It can be assumed that both liquid water resulting from hailstones melting at the surface and liquid rainwater getting to the soil together with hailstones are very close to freezing level ($T_f = 0$ °C) before infiltrating the topsoil layer. In this case, the quantity of hailstones (in kg m$^{-2}$) melting after getting to the soil surface, $I$, can be estimated as:

$$I = \frac{1}{L_f}\left\{C_{5cm}^{t_1}\left(T_{5cm}^{t_1} - T_{5cm}^{t_2}\right) - C_{water}\left(VSM_{5cm}^{t_2} - VSM_{5cm}^{t_1}\right)\left(T_{5cm}^{t_2} - T_f\right)\right\}\Delta z \tag{4}$$

where $L_f = 3.34 \times 10^5$ J kg$^{-1}$ is the latent heat of fusion. Since a fraction of the rain drops may exceed freezing temperature, $I$ as calculated from Eq. (4) is a low estimate.

### 3.3 Nomenclature

- $T_{5cm}$ (°C): in situ soil temperature at a depth of 5 cm;
- $\Delta T_{5cm}$ (°C 12 min$^{-1}$): $T_{5cm}$ change every 12 min during a rainfall event;
- $T_{5cm}$ change range (°C): maximum minus minimum $T_{5cm}$ during a rainfall event, including 12-minute slots just after and before the rainfall event;
- $\delta T_{5cm}$ (°C): $T_{5cm}$ just after the rainfall event minus $T_{5cm}$ just before the rainfall event ;
- VSM$_{5cm}$ (m$^3$ m$^{-3}$): in situ volumetric soil moisture (VSM) at a depth of 5 cm;
- $\Delta$VSM$_{5cm}$ (m$^3$ m$^{-3}$ 12 min$^{-1}$): VSM$_{5cm}$ change every 12 min during a rainfall event;
- VSM$_{5cm}$ change range (m$^3$ m$^{-3}$): maximum minus minimum VSM$_{5cm}$ during a rainfall event, including 12-minute slots just after and before the rainfall event;
- $\delta$VSM$_{5cm}$ (m$^3$ m$^{-3}$): VSM$_{5cm}$ just after the rainfall event minus VSM$_{5cm}$ just before the rainfall event;
- $\Delta z$ (m): depth of the topsoil layer (0.1 m in this study);
- $T_{ISBA}$ (°C): ISBA soil temperature simulations at a depth of 5 cm;
- $\Delta T_{ISBA}$ (°C 12 min$^{-1}$): $T_{ISBA}$ change every 12 min during a rainfall event;





- $T_{air}$ (ºC): observed ambient air temperature at 2 m;

- $T_{wb}$ (ºC): ambient wet-bulb temperature at 2 m calculated using the Stull (2011) equation;

- $T_{rain}$ (ºC): estimated rain temperature;

- RH (dimensionless): in situ air relative humidity at 2 m;

- $VSM_{sat}$ (m$^3$ m$^{-3}$): VSM at saturation (i.e. the soil porosity);

- $\frac{VSM_{5cm}}{VSM_{sat}}$ (dimensionless): $VSM_{5cm}$ to $VSM_{sat}$ ratio or degree of saturation;

- SOM: soil organic matter;

- $f_{clay}$ , $f_{gravel}$, $f_{min}$, $f_{sand}$, $f_{silt}$, $f_{SOM}$ (m$^3$ m$^{-3}$): volumetric fractions of clay, gravels, soil minerals, sand, silt, and SOM; $C_{5cm}^{t}$, $C_{water}$, $C_{min}$, $C_{SOM}$ (J m$^{-3}$ K$^{-1}$): heat capacity values of the topsoil layer at time $t$, water, soil minerals, and

(SOM);

- O, SO, M, MM: oceanic, semi-oceanic, Mediterranean, Mediterranean-mountain climate conditions.

For the first 8 symbols, subscript 5cm stands for observations made at a depth of 5 cm.

In Table 2, marked rainfall events affecting $T_{5cm}$ are defined with $T_{5cm}$ change range $\geq$ 1 ºC. Intense soil-cooling during a marked rainfall event is defined with minimum $\Delta T_{5cm} \leq$ -1.5 ºC in 12 minutes.

In Table 1, the rescaled number of intense soil-cooling events is calculated as:

$$N_{sR} = \frac{N_s}{(1 - f_s)} \qquad (5)$$

where $N_s$ is the number of intense soil-cooling events for one season at one station, and $f_s$ is the proportion of missing data
for the same season at the same station (see Table S1 and Fig. S2 in the Supplement). The $f_s$ values for each season are estimated using the total missing data proportion for all seasons and the scaled seasonal distribution of the fraction of missing data.

## 4 Results

### 4.1 Identification of soil-cooling rains

The various types of rainfall events considered in this study are listed in Table 2. Their frequency is indicated. After data sorting (step 3 in Table 2), only 5.5 % of the fully documented rainfall events can be considered as marked rainfall events, with accumulated precipitation and changes in $VSM_{5cm}$ larger than 5 mm and 0.05 m$^3$m$^{-3}$, respectively. At the same time, this small fraction of rainfall events contributes as much as 57 % of the accumulated rainfall of all rainfall events. On average, 30 marked rainfall events are observed each year at each station.





Among these marked rainfall events, some have a notable impact on the soil temperature and soil moisture profiles. This is illustrated by Fig. 2, showing soil temperature and soil moisture measured at the PRD station from 21 to 25 August 2015 at depths of 5, 10, 20 and 30 cm. A sharp decrease in soil temperature associated to an increase in soil moisture can be observed around noon of 23 August 2015, along with a rainfall event. The most pronounced impacts of the rain are on the

topsoil variables at a depth of 5 cm, but the whole soil profile is affected by the rain, down to a depth of 30 cm. This clearly shows the effects on soil temperature of a soil-cooling rain. Figure 2 also presents the ISBA numerical simulation of the same soil variables. Since ISBA does not represent the heat exchange caused by the mass movement of rainwater, the simulated topsoil temperature is only driven by the surface energy budget, including the evaporation of rainwater intercepted by the vegetation, and by heat conduction from deeper soil layers. As a result, almost no soil temperature change is

simulated while changes in soil moisture in response to the rain are simulated. In this example, the rainfall event as a whole is represented well by the SAFRAN atmospheric forcing used to force the ISBA model. In SAFRAN, the rainfall event lasts for 14 h, starting at 10:00 UTC and ending at 23:00 UTC. Over this period of time, the accumulated rainfall of SAFRAN rain is 86.4 mm, very close to the observed accumulated rainfall of in situ rain of 86.1 mm (note that this time period differs from the one used in Table 4). However, SAFRAN is not able to represent the sub-hourly variability of rainfall intensity.

While the observed peak rainfall intensity value is 27.8 mm in 12 minutes at 12:36 UTC, SAFRAN indicates a rather constant intensity of about 6 mm h$^{-1}$.

The PRD station considered in Fig. 2 is characterized by a Mediterranean climate. The 21 stations of the SMOSMANIA network cover contrasting climate areas (Sect. 2) presenting a different seasonal distribution of precipitation. Figure 3 presents the average monthly rainfall amount across seasons (winter is from December to February) for each SMOSMANIA

station over a 9-year period of time, from 2008 to 2016. Differences in the seasonal distribution of precipitation are very large. For the 9 westernmost stations (from SBR to SFL) the average monthly rainfall amount across seasons is rather homogenous, although stations under oceanic (O) climate tend to have more precipitation in winter and those under semi-oceanic (SO) climate in spring. For the other 12 stations (from MTM to CBR) under Mediterranean (M) and Mediterranean-mountain (MM) climate conditions, summer is generally drier than other seasons. On the other hand, the autumn is wetter

than other seasons, especially in MM climate conditions. The maximum seasonal monthly mean precipitation rate is 272 mm month$^{-1}$ at the BRN station, and LGC, MZN, and BRZ stations, also under MM climate conditions, present values larger than 150 mm month$^{-1}$. The differences in rain intensity distribution are analyzed further in Fig. 4 for marked rainfall events (step 3 in Table 2). We can see that both accumulated rainfall and rain duration of individual marked rainfall events can be much larger for M and MM stations than for O and SO stations, especially in the autumn and in winter. The longest rain duration is

41 hours in winter, and the maximum accumulated rainfall during a single rainfall event is 370 mm in the autumn, at the same BRN station. On the other hand, most of the marked rainfall events of O and SO stations present less than 50 mm accumulated rainfall and last less than 12 h.

The statistical distribution of $\delta T_{5cm}$ ($T_{5cm}$ increase or decrease in topsoil temperature corresponding to a rainfall event) and the $T_{5cm}$ change range for marked rainfall events are shown in Fig. 5. In all climate conditions, the topsoil layer is cooler after





a marked rainfall event with a probability of 80 %. The $\delta T_{5cm}$ difference values are larger than 1 °C or smaller than -1 °C with a probability of 25 %, only. More often than not, a cooling is observed in these conditions, rather than a warming. The probability of the $T_{5cm}$ change range to equal or exceed 1 °C is a bit larger: 28 %. This criterion was used to select marked rainfall events affecting $T_{5cm}$ (step 4 in Table 2). After data sorting, we obtain a total of 1577 events. This corresponds to

about 8 events per year and per station. Because a lot of marked rainfall events can last several hours, $T_{5cm}$ change range values $\geq$ 1 °C can be explained by the diurnal cycle of the surface net radiation, rather than to the mass movement of rainwater. Since obvious soil-warming rainfall events are not detectable in our observations, we focus on soil-cooling events characterized by a sharp decrease of topsoil temperature (e.g. in Fig. 2) during the 12-min time interval of the soil profile observations. For this purpose, step 5 in Table 2 permits selecting 122 intense soil-cooling rains using minimum $\Delta T_{5cm}$

values $\leq$ -1.5 °C in 12 minutes. This corresponds to 0.65 events per year and per station. Figure 6 presents the statistical distribution of minimum $\Delta T_{5cm}$ observations, and the corresponding minimum $\Delta T_{5cm}$ values simulated by the ISBA model for the 1577 marked rainfall events affecting $T_{5cm}$. It appears that step 5 tends to remove the longest rainfall events and the selected rainfall events last less than 4 hours. The comparison between observed and simulated values shows that ISBA is not able to simulate $\Delta T_{5cm}$ values well. In particular, most of the simulated minimum $\Delta T_{5cm}$ values are larger than -0.5 °C in

12 minutes during intense soil-cooling events, even for very intense ones with observed minimum $\Delta T_{5cm}$ values lower than -4 °C in 12 minutes. Figure 6 also shows that most of the 122 intense soil-cooling events occur in M or in MM climate conditions.

## 4.2 Frequency of intense soil-cooling rains

Among the 122 identified intense soil-cooling events, 107 occur at stations under M or MM climates, only 15 under the O or

SO climates. The spatial and seasonal distribution of these intense soil-cooling rainfall events is shown in Fig. 7 for each station of the SMOSMANIA network. Most of the intense soil-cooling rains (82) are in summer, while only 4 are found in winter. The latter are all for the same BRZ station, under MM climate conditions. In spring and during the autumn, 17 and 19 events are observed, respectively. At 6 stations, no intense soil-cooling rain is observed in 9 years, 3 are under O climate conditions (SBR, URG, CRD), 2 under SO climate conditions (CDM, MNT), and 1 under MM climate conditions (MZN).

The PRD and BRZ stations, under M and MM climate conditions, respectively, present the largest number of events, with a mean rescaled frequency of intense soil-cooling rains of 2.7 per year (Table 1). For the 12 M or MM stations (from MTM to CBR) the mean frequency of intense soil-cooling rains is once a year. More details are shown in Table 1.

More characteristics of these 122 intense soil-cooling rains are shown in Fig. 8. These events do not always correspond to a large amount of accumulated rainfall. Actually, about 80 % of these events present accumulated rainfall values smaller than

30 mm. About 80 % of these rains last less than 2 hours, and the longest rain duration is less than 5 hours. The mean hourly rain rate per event does not exceed 30 mm h$^{-1}$ for 90 % of the events. This shows that extremely large amounts of rain or large precipitation intensity are not required to produce intense soil-cooling. Figure 8 also shows that while 82 % of intense soil-cooling rains present minimum $\Delta T_{5cm}$ values larger than -3 °C 12 min$^{-1}$, very low values (down to -6.5 °C 12 min$^{-1}$) can



be observed. During intense soil-cooling rains, the minimum $\Delta T_{5cm}$ contributes to increase the $T_{5cm}$ change range. For 18 % of the events, the minimum $\Delta T_{5cm}$ represents more than 80 % of the $T_{5cm}$ change range. For 48 % of the events, the minimum $\Delta T_{5cm}$ represents more than 60 % of the $T_{5cm}$ change range. The statistical distributions of $\delta T_{5cm}$ and $\delta VSM_{5cm}$ are also shown in Fig. 8. For intense soil-cooling rains, $T_{5cm}$ is always lower after the rain than before. The major part (74 %) of the $\delta T_{5cm}$ values ranges between -6 °C and -2 °C. For $\delta VSM_{5cm}$, 8 % of the values are slightly negative (the minimum observed $\delta VSM_{5cm}$ is -0.006 m$^3$ m$^{-3}$), and 21 % do not exceed 0.050 m$^3$ m$^{-3}$. The maximum observed $\delta VSM_{5cm}$ is 0.3 m$^3$ m$^{-3}$.

Figure 9 investigates the distribution of the starting time of the intense soil-cooling rains at a resolution of 1 hour together with the corresponding average rain rate (in mm h$^{-1}$) across seasons. For the 4 winter events, the rain rate is relative small (less than 7.5 mm h$^{-1}$). Only 3 events are found with a rain rate larger than 50 mm h$^{-1}$, two in summer and one during the autumn. It can be seen that the intense soil-cooling rains tend to occur at daytime. A large proportion (83 %) of the 122 intense soil-cooling rains occur between 09:00 and 21:00 UTC, which is much larger than the fraction of 67 % observed for the 1577 marked rainfall events affecting $T_{5cm}$. The intense soil-cooling rains are rather uniformly distributed between 09:00 and 21:00 UTC.

## 4.3 Estimation of rain temperature from soil temperature and soil moisture observations

Among the 122 intense soil-cooling events, we found 13 cases presenting simultaneous marked changes in $T_{5cm}$, $VSM_{5cm}$, and $VSM_{10cm}$, within a 12-minute slot (step 6 in Table 2). Observed values of $VSM_{5cm}$, $T_{5cm}$, 2 m air temperature $T_{air}$ at time $t_1$ and $t_2$, and 2 m wet-bub temperature $T_{wb}$ at time $t_1$ for these 13 example rains are listed in Table 4, together with the accumulated precipitation of the considered rainfall events and the peak rainfall intensity (mm 12 min$^{-1}$) of each rain. Among these 13 cases, 2 are under SO climate conditions (PRG and SFL), and 11 are under M and MM climate conditions (from LZC to CBR in Table 4). The latter include 5 cases from the same station, PRD. From a seasonal perspective, 2 cases occurred in spring (cases 9 and 12), one during the autumn (case 3), and the other 10 cases occurred in summer.

Table 4 also shows the rain temperature estimates at time $t_1$ ($T_{rain}$) derived from Eq. (3), and the amount of melted hail derived from Eq. (4). Interestingly, the only two cases occurring in spring (cases 9 and 12) correspond to melting hail events. Taking the PRD station case 9 as an example, the measured precipitation amount is 9.3 kg m$^{-2}$ during a rainfall event of 36 min, with an average rain rate of 15.5 mm h$^{-1}$. From time $t_1$ to $t_2$, the $T_{5cm}$ topsoil temperature decreases very fast from 17.9 to 12.6 °C (-5.3 °C in 12 min), and the air temperature also decreases from 17.5 to 15.0 °C (-2.5 °C in 12 min). During the same time lapse, $VSM_{5cm}$ increases by +0.13 m$^3$ m$^{-3}$. Storms with hail were reported in the press and in social media at many places of southern France on 23 April 2016, including close to the PRD region (Infoclimat, 2016). Using Eq. (4), one can estimate the amount of water originating from melting hail: about 1 kg m$^{-2}$. Soil temperature and soil moisture time series for all these 13 rain examples are shown in Figs. S3 - S15. Thunderstorms and hail were also reported for case 12 (Infoclimat, 2010).





Figure 10 shows the estimated $T_{rain}$ vs. $T_{air}$ and $T_{5cm}$ at time $t_1$. While most of the $T_{air}$ values range between 16 and 22 °C, except for case 12 ($T_{air} = 4.3$ °C), the estimated $T_{rain}$ present a larger variability, from 0 to 22.5 °C. Excluding the two spring cases (9 and 12), the standard deviation of $T_{rain}$ values in Table 4 is 4.6 °C, much larger than for $T_{air}$, 1.9°C.

5 For the 13 storms listed in Table 4, $T_{rain}$ ($T_{5cm}$) tends to be lower (higher) than $T_{air}$, with a mean difference of -5.1 °C (+6.0 °C). On average, $T_{rain}$ is cooler than topsoil by -11.1 °C. $T_{rain}$ is cooler than $T_{wb}$ by -3.8 °C. For cases 2, 11, and 13, $T_{rain}$ is close to $T_{air}$. For cases 1 and 3, $T_{rain}$ is close to $T_{wb}$. The other cases present $T_{rain}$ values much cooler than $T_{air}$ and $T_{wb}$. In particular, $T_{rain}$ is cooler than $T_{air}$ by more than 5 °C for 5 cases (4, 5, 7, 9, 10), among which 3 cases (7, 9, 10) occurred at the PRD station. At time $t_1$, RH ranges from 68 % to 97 % and $\frac{VSM_{5cm}}{VSM_{sat}}$ ranges from 20.7 % to 63.8 %. Soil-cooling rate $\Delta T_{5cm}$ values range from -5.3 °C to -1.5 °C in 12-min. The corresponding air cooling values are less pronounced, ranging from -2.5

10 °C to 0.0 °C. For case 9, $\frac{VSM_{5cm}}{VSM_{sat}}$ increases only by 27 % during the considered 12-min slot. This is a relative small increase compared to other cases, less than the median value of 29 % and much less than the maximum value of 58 % observed for case 10 at the same station. Despite the moderate soil wetting in case 9, $T_{5cm}$ values presented the most pronounced decrease (-5.3 °C). The $T_{5cm}$ value at time $t_2$ also presented the largest difference with the estimated rain temperature (+12.6 °C).

## 5 Discussion

### 5.1 How accurate are rain temperature estimates?

In this study, an attempt is made to estimate rain temperature using observations in the topsoil layer. Because Eqs. (3) and (4) include soil heat capacity (Eq. (1)), the static soil properties must be known together with the time-evolving VSM and soil temperature. In particular, the volumetric fraction of gravels is the most variable static soil characteristic in Table 3: $f_{gravel}$ ranges from 0 to 0.41 m$^3$ m$^{-3}$ at CRD and BRN stations, respectively. Among stations of the 13 rain retrieval cases in

Table 4, $f_{gravel}$ ranges from 0.05 to 0.34 m$^3$ m$^{-3}$ at PZN and PRD, respectively. The fraction of gravels was not measured at a depth of -5 cm. Instead, values given in Table 3 are derived from gravimetric measurements made at -10 cm. In order to assess to what extent uncertainties on $f_{min}$ values may affect the retrieved $T_{rain}$ values, two numerical experiments (Exp1 and Exp2) were made using other soil characteristics than those listed in Table 3 (Control experiment):

- Exp1 used the reassembled static soil volumetric fractions of soil minerals and SOM at a depth of 5 cm assuming
$f_{gravel} = 0$ m$^3$ m$^{-3}$. This was equivalent to considering fine earth only and the resulting $f_{SOM}$ and $f_{min}$ fractions were larger and smaller than in the Control experiment, respectively. This tended to increase $C_{5cm}^t$ (Eq. (1)), and to decrease $T_{rain}$ (Eq. (3)).

- Exp2 used the measured soil characteristics at a depth of 10 cm (Calvet et al., 2016). The impact of Exp2 on $f_{SOM}$, $f_{min}$, $C_{5cm}^t$, and $T_{rain}$ varied a lot from one station to another.

Volumetric fractions of the topsoil elements used in Exp1 and Exp2 are listed in Tables S2 and S3, respectively. Differences in $f_{SOM}$ and $f_{min}$ values are listed in Table 5.



Differences in estimated $T_{rain}$ values for Exp1 and Exp2 with respect to the Control are shown in Table 5. In Exp1, $T_{rain}$ estimates tend to present slightly lower values, with differences down to -0.3 °C, and the median difference value is -0.1 °C, with a standard deviation of 0.1 °C. In Exp2, $T_{rain}$ differences range from -1.23 °C to +0.17 °C, and the median difference value is 0 °C, with a standard deviation of 0.4 °C. Merging results from Exp1 and Exp2, 80 % of the statistical distribution of $T_{rain}$ differences range between -0.3 °C and +0.1 °C. The most marked changes in $T_{rain}$ (-0.50 °C and -1.23 °C) are observed for Exp2 (cases 1 and 4, at PRG and NBN stations, respectively). They correspond to the largest changes in $f_{min}$ (+0.070 and +0.105, respectively). This gives an idea of the uncertainties on $T_{rain}$ related to poorly known soil heterogeneities and to their impact on the soil characteristics measured in the field.

Another source of uncertainties is that the topsoil layer, from the soil surface down to a depth of -0.1 m, may not be completely affected by the rainfall event, or that the instruments positioned at a depth of -5 cm may not be able to sample mean values relevant for the topsoil layer. In order to limit this effect, we selected only 13 intense soil-cooling events by imposing a marked change in $VSM_{10cm}$ during the considered 12-min slot. If the latter condition is ignored, 32 soil-cooling events can be considered instead of 13, and we checked that similar results are found (not shown).

A limitation of the method used in this study is that the soil moisture and soil temperature probes at a depth of 5 cm are not placed at exactly the same location. We found some examples for which the VSM response to rain does not match with the drop in topsoil temperature (Figs. S18, S19, S20, S21 and S22). This limited the number of events for which rainwater temperature could be estimated.

More research is needed to develop techniques to measure rainwater temperature. Instruments similar to the rain-temperature equipment of Byers et al. (1949) could be developed. Our results show that using automatic temperature and volumetric moisture observations in a porous medium of known thermal properties has potential to estimate $T_{rain}$ and possibly the amount of hailstones in real time.

### 5.2 Does soil-cooling matter?

We showed (e.g. Fig. 2) that the temperature of rain drops reaching the soil surface can impact the soil temperature profile during several hours. Investigating the impact on longer time periods would require using a LSM able to activate or deactivate the representation of sensible heat input from liquid water into the soil. This impact was investigated experimentally by Wierenga et al. (1975) through an irrigation experiment with cold and warm water (4.1 °C and 21.6 °C, respectively). They showed that 42 hours were needed before differences in soil temperature at a depth of 0.2 m were reduced to less than 1 °C, and more than 5 days below a depth of 0.5 m. They used quite large irrigation amounts of more than 120 mm. Figure 8 shows that accumulated rainfall during one event can exceed 120 mm in Mediterranean climate conditions (M and MM). Such events are not observed in O and SO conditions but less intense precipitation events can impact the surface energy budget, even if this is not obvious in the soil temperature time series. Figure 8 shows that intense soil-cooling events (step 5 in Table 2) are associated to rather uniformly distributed increases in topsoil VSM values. Actually, the statistical distribution of $\delta VSM_{5cm}$ values tends to shift towards larger values at each data sorting step listed in





Table 2. This is illustrated by Fig. 11 for steps 2, 3, and 4. For fully documented rainfall events (step 2), $VSM_{5cm}$ does not increase for about 60 % of the rainfall events ($\delta VSM_{5cm} \leq 0$ m$^3$ m$^{-3}$ 12 min$^{-1}$). Two possible reasons are that (1) rainwater can be intercepted by vegetation and/or litter, especially when the rain is very slight and/or the soil ground is very dry, (2) $VSM_{5cm}$ is close to saturation, so as to no $VSM_{5cm}$ increase is observed. We found examples under the above two situations, shown in Fig. S16 ($VSM_{5cm}$ is relatively small and the rainwater might be intercepted) and Fig. S17 ($VSM_{5cm}$ is close to saturation). For steps 3 and 4, Fig. 11 shows that only 7 % to 9 % of $VSM_{5cm}$ values do not increase. About the same proportion is observed for the 122 intense soil-cooling rains (Fig. 8).

Developing LSMs able to represent the sensible heat input from liquid water into the soil is needed, as well as a way to diagnose rainwater temperature from atmospheric model simulations (Feiccabrino et al., 2015). Attempts were made in a few studies to represent this effect (e.g. Emanuel et al., 2008; Wang et al., 2016) but in general, it was assumed that $T_{rain}$ was equal to $T_{wb}$. This study shows that $T_{rain}$ can be much lower than $T_{wb}$ during severe convective events and confirms the findings of Byers et al. (1949). Since severe convective events associated to the intense soil-cooling events observed in this study tend to become more and more frequent in relation to climate change (Feng et al., 2016), soil-cooling effects may play a role in the response of the Earth system to climate change. Moreover, rainwater temperature estimates from observation networks or from atmospheric model simulations could be beneficial for a number of applications such as urban heat island monitoring (e.g. Jelinkova et al., 2015), drink-water quality monitoring (e.g. Chubaka et al., 2018), the estimation of the emission rates of greenhouse gases by soils (e.g. Gagnon et al., 2018), or the quantification of soil erosion (e.g. Sachs and Sarah, 2017).

## 6 Conclusions

In situ rain-temperature measurements are rare. We used the soil moisture and soil temperature observations from the SMOSMANIA network over 9 years in southern France to assess the cooling effects on soils of rainfall events. The rainwater temperature was estimated using observed changes of topsoil volumetric soil moisture and soil temperature in response to the rainfall event. We found that most (72 %) marked rainfall events did not impact $T_{5cm}$ change range more than ±1 ℃. On the other hand, about 2 % of marked rainfall events triggered intense soil-cooling with drops in $\Delta T_{5cm} \leq$ -1.5 ℃ in 12 min. Such intense soil-cooling rains were mainly observed under Mediterranean climate conditions, in summer, at daytime. The average frequency of the occurrence of such events for the 12 Mediterranean stations was once a year. Among all these intense soil-cooling rains, the minimum observed value of $\Delta T_{5cm}$ was -6.5 ℃ in 12 min. Rain temperature estimates were obtained for 13 cases. They were generally lower than the ambient air temperatures, wet-bulb temperatures, and $T_{5cm}$ values (with mean differences of -5.1, -3.8, and -11.1 ℃, respectively). In 5 cases, rain temperature estimates were much cooler than air temperature, by at least -5 ℃ and down to -17.5 ℃, likely in relation to hailstones melting just before reaching the surface or melting at the surface of the soil. More research is needed to develop measurement techniques for rainwater temperature and perform such measurements in contrasting climate conditions.



*Data availability.*

The soil moisture (temperature) observations are available to the research community through the International Soil Moisture Network website (ISMN, 2018).

*Acknowledgments.*

This work is a contribution to the HyMeX program (https://www.hymex.org/). We thank our Météo-France colleagues for their support in collecting, checking and archiving the SMOSMANIA data: Annick Auffray, Catherine Bienaimé, Marc Bailleul, Laurent Brunier, Jérôme Candiago, Anne Chaumont, Jacques Couzinier, Mathieu Créau, Hélène Fillancq, Philippe
Gillodes, Sandrine Girres, Michel Gouverneur, Didier Grimal, Viviane Isler, Maryvonne Kerdoncuff, Matthieu Lacan, Pierre Lantuejoul, William Maurel, Roland Mazurie, Dominique Paulais, Bruno Piguet, Fabienne Simon, Dominique Simonpietri, Marie-Hélène Théron and Marie Yardin.

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



**Table 1.** The environment characteristics for the 21 stations of the SMOSMANIA network and the frequency of intense soil-cooling during marked rainfall events from 2008 to 2016. The number of intense soil-cooling rains, the frequency and the mean time lag between two intense soil-cooling rains are rescaled according to the fraction of missing data across seasons (see Table S1). Stations are listed from (top) west to (bottom) east.

| Station | Altitude (m) | Climate | Soil type | Number of intense soil-cooling events | Rescaled number of intense soil-cooling events | Rescaled frequency of intense soil-cooling events (year$^{-1}$) | Rescaled mean time lag between two intense soil-cooling events (months) |
|---|---|---|---|---|---|---|---|
| SBR | 81 | Oceanic | Sand | 0 | 0.0 | 0.0 | - |
| URG | 145 | Oceanic | Silt loam | 0 | 0.0 | 0.0 | - |
| CRD | 149 | Oceanic | Sand | 0 | 0.0 | 0.0 | - |
| PRG | 245 | Semi-oceanic | Silty clay | 2 | 2.0 | 0.2 | 54.0 |
| CDM | 174 | Semi-oceanic | Silty clay | 0 | 0.0 | 0.0 | - |
| LHS | 249 | Semi-oceanic | Clay loam | 6 | 6.2 | 0.7 | 17.4 |
| SVN | 158 | Semi-oceanic | Loam | 6 | 6.3 | 0.7 | 17.1 |
| MNT | 295 | Semi-oceanic | Silt loam | 0 | 0.0 | 0.0 | - |
| SFL | 330 | Semi-oceanic | Loam | 1 | 1.1 | 0.1 | 98.2 |
| MTM | 538 | Mediterranean/Mountain | Clay loam | 1 | 1.0 | 0.1 | 108.0 |
| LZC | 60 | Mediterranean | Sandy clay loam | 6 | 6.3 | 0.7 | 17.1 |
| NBN | 112 | Mediterranean | Clay | 4 | 4.1 | 0.5 | 26.3 |
| PZN | 30 | Mediterranean | Sandy loam | 3 | 3.1 | 0.3 | 34.8 |
| PRD | 85 | Mediterranean | Clay loam | 23 | 24.3 | 2.7 | 4.4 |
| LGC | 499 | Mediterranean/Mountain | Loamy sand | 11 | 11.4 | 1.3 | 9.5 |
| MZN | 1240 | Mediterranean/Mountain | Sandy loam | 0 | 0.0 | 0.0 | - |
| VLV | 41 | Mediterranean | Sandy loam | 3 | 3.4 | 0.4 | 31.8 |
| BRN | 480 | Mediterranean/Mountain | Loamy sand | 11 | 11.7 | 1.3 | 9.2 |
| MJN | 318 | Mediterranean | Loam | 15 | 16.3 | 1.8 | 6.6 |
| BRZ | 650 | Mediterranean/Mountain | Clay loam | 22 | 24.0 | 2.7 | 4.5 |
| CBR | 142 | Mediterranean | Sandy clay loam | 8 | 8.3 | 0.9 | 13.0 |



**Table 2.** Steps for identifying marked soil-cooling rains, together with the number of events for all the SMOSMANIA stations from 2008 to 2016.

| Step | Event to be identified | Total number of events (and number per year and per station) | Definition |
|---|---|---|---|
| 1 | Rainfall event | 123215 (652) | Continuous time series of non-zero accumulated liquid precipitation values at time intervals of 12 minutes |
| 2 | Fully documented rainfall event | 104178 (551) | Rainfall event with complete in situ $T_{5cm}$ and $VSM_{5cm}$ observation time series |
| 3 | Marked rainfall event | 5714 (30) | Fully documented rainfall event with accumulated precipitation and changes in $VSM_{5cm}$ larger than 5 mm and equal to or above 0.05 $m^3$ $m^{-3}$ ($VSM_{5cm}$ change range $\geq$ 0.05 $m^3$ $m^{-3}$), respectively |
| 4 | Marked rainfall event affecting $T_{5cm}$ topsoil temperature | 1577 (8) | Marked rainfall event with changes in $T_{5cm}$ equal to or above 1 ℃ ($T_{5cm}$ change range $\geq$ 1 ℃) |
| 5 | Intense soil-cooling during a marked rainfall event | 122 (0.65) | A 12-minute slot during a marked rainfall event affecting $T_{5cm}$ with minimum $\Delta T_{5cm}$ (drop of $T_{5cm}$ within 12 minutes) equal to or below -1.5 ℃ |
| 6 | Rainwater temperature retrieval slot | 13 (0.07) | Intense soil-cooling during a marked rainfall event with rises in $VSM_{5cm}$ and $VSM_{10cm}$ equal to or above 0.10 and 0.05 $m^3$ $m^{-3}$, respectively |





**Table 3.** Soil characteristics at a depth of 5 cm for the 21 stations of the SMOSMANIA network. The volumetric fractions of sand, clay and silt at 5 cm ($f_{sand}$, $f_{clay}$ and $f_{silt}$) are measured, and the volumetric fractions of gravel and soil organic matter (SOM) ($f_{gravel}$ and $f_{SOM}$) at 5 cm are derived from measurements at 10 cm. VSM$_{sat}$ is the porosity representing VSM at saturation (VSM$_{sat}$ = 1 - $f_{sand}$ - $f_{clay}$ - $f_{silt}$ - $f_{gravel}$ - $f_{SOM}$ ). Stations are listed from (top) west to (bottom) east.

| Station | VSM$_{sat}$ (m$^3$ m$^{-3}$) | $f_{sand}$ (m$^3$ m$^{-3}$) | $f_{clay}$ (m$^3$ m$^{-3}$) | $f_{silt}$ (m$^3$ m$^{-3}$) | $f_{gravel}$ (m$^3$ m$^{-3}$) | $f_{SOM}$ (m$^3$ m$^{-3}$) |
|---|---|---|---|---|---|---|
| SBR | 0.440 | 0.501 | 0.021 | 0.016 | 0.001 | 0.021 |
| URG | 0.472 | 0.079 | 0.078 | 0.338 | 0.005 | 0.029 |
| CRD | 0.487 | 0.413 | 0.027 | 0.029 | 0.000 | 0.045 |
| PRG | 0.505 | 0.045 | 0.119 | 0.121 | 0.186 | 0.024 |
| CDM | 0.457 | 0.076 | 0.211 | 0.227 | 0.011 | 0.018 |
| LHS | 0.415 | 0.140 | 0.178 | 0.186 | 0.052 | 0.029 |
| SVN | 0.459 | 0.134 | 0.072 | 0.166 | 0.159 | 0.010 |
| MNT | 0.457 | 0.117 | 0.062 | 0.235 | 0.099 | 0.029 |
| SFL | 0.401 | 0.143 | 0.075 | 0.111 | 0.257 | 0.013 |
| MTM | 0.440 | 0.103 | 0.072 | 0.070 | 0.277 | 0.038 |
| LZC | 0.472 | 0.107 | 0.066 | 0.070 | 0.270 | 0.015 |
| NBN | 0.511 | 0.058 | 0.103 | 0.061 | 0.236 | 0.030 |
| PZN | 0.513 | 0.199 | 0.069 | 0.126 | 0.051 | 0.042 |
| PRD | 0.479 | 0.044 | 0.051 | 0.069 | 0.336 | 0.020 |
| LGC | 0.411 | 0.254 | 0.045 | 0.048 | 0.220 | 0.022 |
| MZN | 0.609 | 0.168 | 0.037 | 0.043 | 0.085 | 0.058 |
| VLV | 0.559 | 0.244 | 0.051 | 0.077 | 0.027 | 0.043 |
| BRN | 0.455 | 0.087 | 0.011 | 0.015 | 0.415 | 0.018 |
| MJN | 0.500 | 0.065 | 0.023 | 0.055 | 0.317 | 0.040 |
| BRZ | 0.579 | 0.090 | 0.056 | 0.094 | 0.163 | 0.019 |
| CBR | 0.500 | 0.113 | 0.058 | 0.067 | 0.239 | 0.023 |



**Table 4.** The estimated rain temperatures ($T_{\text{rain}}$) for 13 intense soil-cooling events, together with the in situ observations of $VSM_{5cm}$, $T_{5cm}$, $T_{\text{air}}$ at time $t_1$ and $t_2$, and $T_{\text{wb}}$ at time $t_1$. The time lapse from time $t_1$ to $t_2$ is 12 min. For rain events 9 and 12, Eq. (4) is used to estimate the amount of melting hail at the soil surface. $T_{\text{rain}}$ values lower by -5 °C than $T_{\text{air}}$ at time $t_1$ are in bold. Stations are listed from west (top) to east (bottom).

| Station | Rain event number in Fig. 10 | Date (year-month-day) | Accumulated rainfall (mm event$^{-1}$) | Peak rainfall intensity (mm 12 min$^{-1}$) | $VSM_{5cm}$ at $t_1$ (m$^3$ m$^{-3}$) | $VSM_{5cm}$ at $t_2$ (m$^3$ m$^{-3}$) | $T_{5cm}$ at $t_1$ (°C) | $T_{5cm}$ at $t_2$ (°C) | $T_{wb}$ at $t_1$ (°C) | $T_{air}$ at $t_1$ (°C) | $T_{air}$ at $t_2$ (°C) | $T_{rain}$ at $t_1$ (°C) | Melting hail (kg m$^{-2}$) |
|---|---|---|---|---|---|---|---|---|---|---|---|---|---|
| PRG | 1 | 2008-07-26 | 28.1 | 16.5 | 0.232 | 0.339 | 21.4 | 19.9 | 14.5 | 16.2 | 16.2 | 13.3 | 0 |
| SFL | 2 | 2015-08-31 | 29.0 | 16.1 | 0.132 | 0.310 | 26.3 | 24.1 | 16.2 | 17.2 | 16.4 | 18.9 | 0 |
| LZC | 3 | 2009-10-08 | 10.0 | 6.6 | 0.127 | 0.262 | 24.0 | 21.8 | 15.3 | 16.0 | 14.3 | 15.6 | 0 |
| NBN | 4 | 2011-08-14 | 14.4 | 10.0 | 0.141 | 0.263 | 26.3 | 23.5 | 18.5 | 19.9 | 19.3 | **14.8** | 0 |
| PZN | 5 | 2015-08-31 | 29.0 | 9.6 | 0.166 | 0.349 | 23.6 | 19.6 | 18.2 | 18.7 | 18.6 | **10.8** | 0 |
| PRD | 6 | 2011-07-13 | 26.2 | 20.5 | 0.099 | 0.338 | 25.8 | 22.5 | 19.8 | 20.2 | 19.8 | 17.7 | 0 |
| PRD | 7 | 2015-08-23 | 77.6 | 27.8 | 0.252 | 0.360 | 21.7 | 19.0 | 18.1 | 19.5 | 18.1 | **6.5** | 0 |
| PRD | 8 | 2015-08-31 | 16.9 | 4.8 | 0.161 | 0.274 | 24.4 | 22.7 | 19.6 | 20.7 | 18.8 | 16.5 | 0 |
| PRD | 9 | 2016-04-13 | 9.3 | 8.5 | 0.212 | 0.340 | 17.9 | 12.6 | 13.7 | 17.5 | 15.0 | **0** | 1.1 |
| PRD | 10 | 2016-06-15 | 24.1 | 20.7 | 0.126 | 0.402 | 22.6 | 18.0 | 17.2 | 17.6 | 16.2 | **11.7** | 0 |
| LGC | 11 | 2016-06-07 | 13.9 | 6.5 | 0.117 | 0.247 | 27.5 | 25.7 | 17.5 | 19.9 | 17.7 | 20.2 | 0 |
| MJN | 12 | 2010-03-30 | 20.3 | 5.6 | 0.319 | 0.442 | 9.8 | 7.7 | 3.4 | 4.3 | 2.7 | **0** | 0.3 |
| CBR | 13 | 2010-06-30 | 41.1 | 16.9 | 0.170 | 0.360 | 31.4 | 28.6 | 19.0 | 21.9 | 20.7 | 22.5 | 0 |




**Table 5.** The estimated $T_{rain}$ differences, $f_{min}$ differences and $f_{SOM}$ differences between Exp1 and Control, and between Exp2 and Control. Changes in values of $T_{rain}$ larger than ±0.5 °C are in bold. Changes in values of volumetric fractions larger than 0.05 $m^3 m^{-3}$ are in bold.

| Station | Rain event number in Fig. 10 | $T_{rain}$ (°C) | | $f_{min}$ ($m^3 m^{-3}$) × 100 | | $f_{SOM}$ ($m^3 m^{-3}$) × 100 | |
|---|---|---|---|---|---|---|---|
| | | Exp1 - Control | Exp2 - Control | Exp1 - Control | Exp2 - Control | Exp1 - Control | Exp2 - Control |
| PRG | 1 | -0.07 | **-0.50** | -0.8 | **7.0** | 1.5 | 0.4 |
| SFL | 2 | -0.06 | 0.05 | -0.5 | -2.0 | 1.1 | 0.8 |
| LZC | 3 | -0.09 | -0.32 | -0.8 | 4.2 | 1.6 | 0.0 |
| NBN | 4 | -0.26 | **-1.23** | -1.4 | **10.5** | 3.0 | 0.6 |
| PZN | 5 | -0.04 | -0.14 | -0.3 | 3.7 | 0.5 | -1.9 |
| PRD | 6 | -0.19 | 0.09 | -1.8 | -1.5 | 3.7 | 0.1 |
| PRD | 7 | -0.33 | 0.17 | -1.8 | -1.5 | 3.7 | 0.1 |
| PRD | 8 | -0.21 | 0.10 | -1.8 | -1.5 | 3.7 | 0.1 |
| PRD | 9 | 0 | 0 | -1.8 | -1.5 | 3.7 | 0.1 |
| PRD | 10 | -0.22 | 0.11 | -1.8 | -1.5 | 3.7 | 0.1 |
| LGC | 11 | -0.06 | 0.11 | -0.7 | -1.3 | 1.3 | -0.3 |
| MJN | 12 | 0 | 0 | -3.9 | 0.6 | **7.9** | -1.2 |
| CBR | 13 | -0.12 | 0.03 | -1.0 | 0.9 | 2.2 | -1.0 |



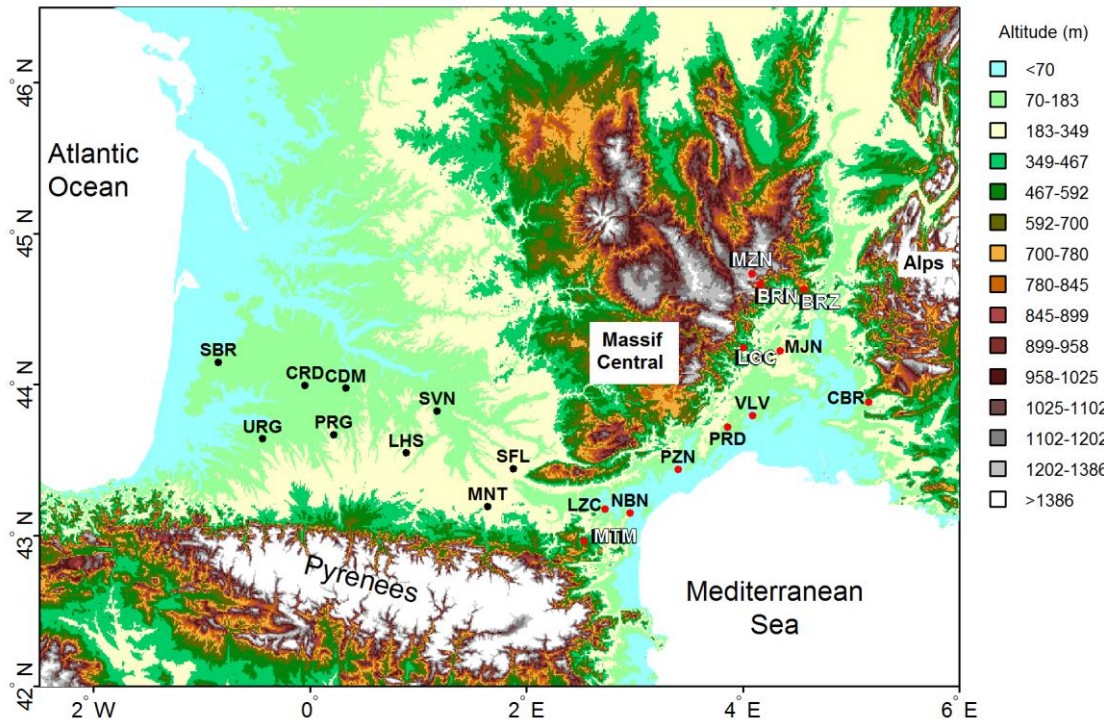

**Figure 1. Locations of the 21 SMOSMANIA stations in southern France. Black dots indicate stations under oceanic and semi-oceanic climates. Red dots indicate stations under Mediterranean and Mediterranean-mountain climates. White letters are for names of the five stations under Mediterranean-mountain climates. Background geographic altitudes are from SRTM 90 m digital elevation data (http://srtm.csi.cgiar.org/).**





**Figure 2. In situ soil temperature (a) and soil moisture (b) measured at the PRD station from 21 to 25 August 2015 at depths of 5, 10, 20 and 30 cm, together with the in situ rainfall observations (mm 12 min$^{-1}$) shown in grey. ISBA simulations of soil temperature (c) and soil moisture (d), together with the SAFRAN rainfall data (mm hour$^{-1}$) shown in grey.**



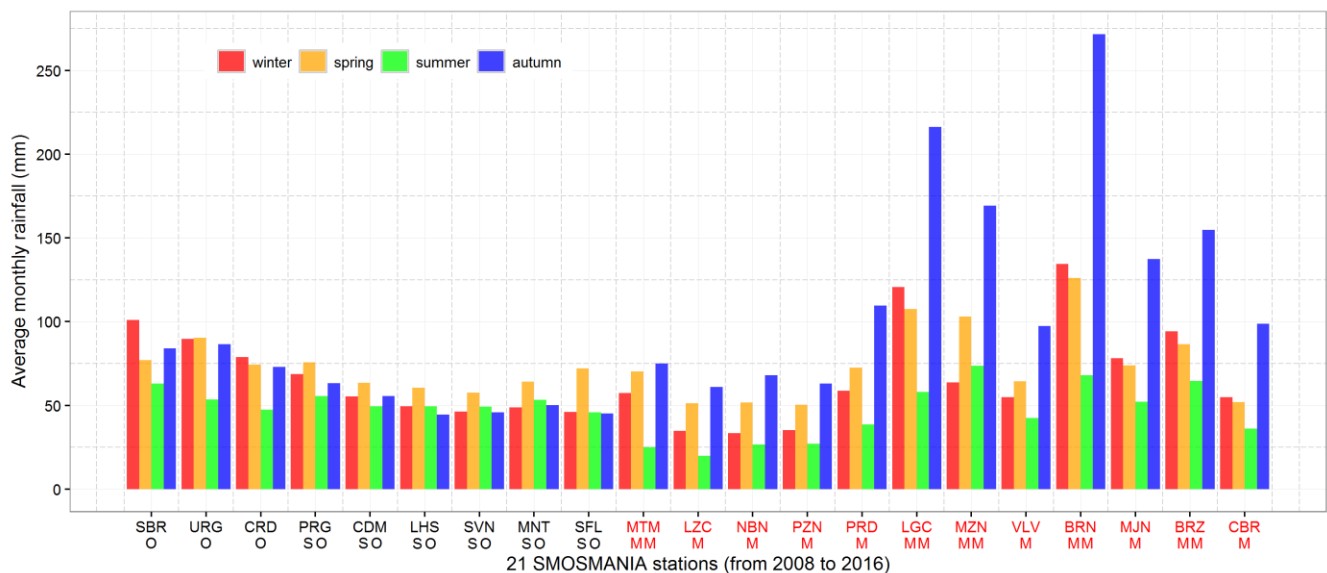

5    **Figure 3.** Average monthly rainfall (in units of mm) for the 21 SMOSMANIA stations across seasons from 2008 to 2016. Stations are sorted from (left) west to (right) east. Symbols "O", "SO", "M", and "MM" indicate Oceanic, Semi-Oceanic, Mediterranean, and Mediterranean-mountain climates, respectively. Winter season is from December to February.



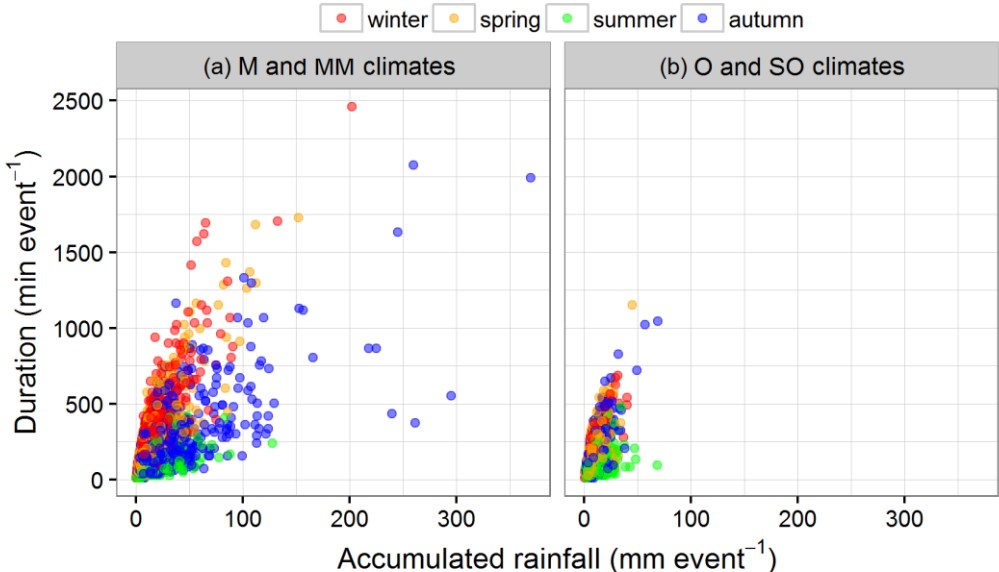

**Figure 4. Rainfall duration vs. accumulated rainfall across seasons for each marked rainfall event at Mediterranean (M) and Mediterranean-mountain (MM) stations (a), and oceanic (O) and semi-oceanic (SO) stations (b).**



**Figure 5. Statistical distributions of** $\delta T_{5cm}$ **(a, b) and** $T_{5cm}$ **change range (c, d) during 5714 marked rainfall events for Mediterranean (M) and Mediterranean-mountain (MM) stations (a, c) and oceanic (O) and semi-oceanic (SO) stations (b, d). The percent value is the ratio of the count of each bin to the total count in all climate conditions. Bin width is 1 ºC. Bins for values larger than 1 °C or smaller than -1 °C are in red.**





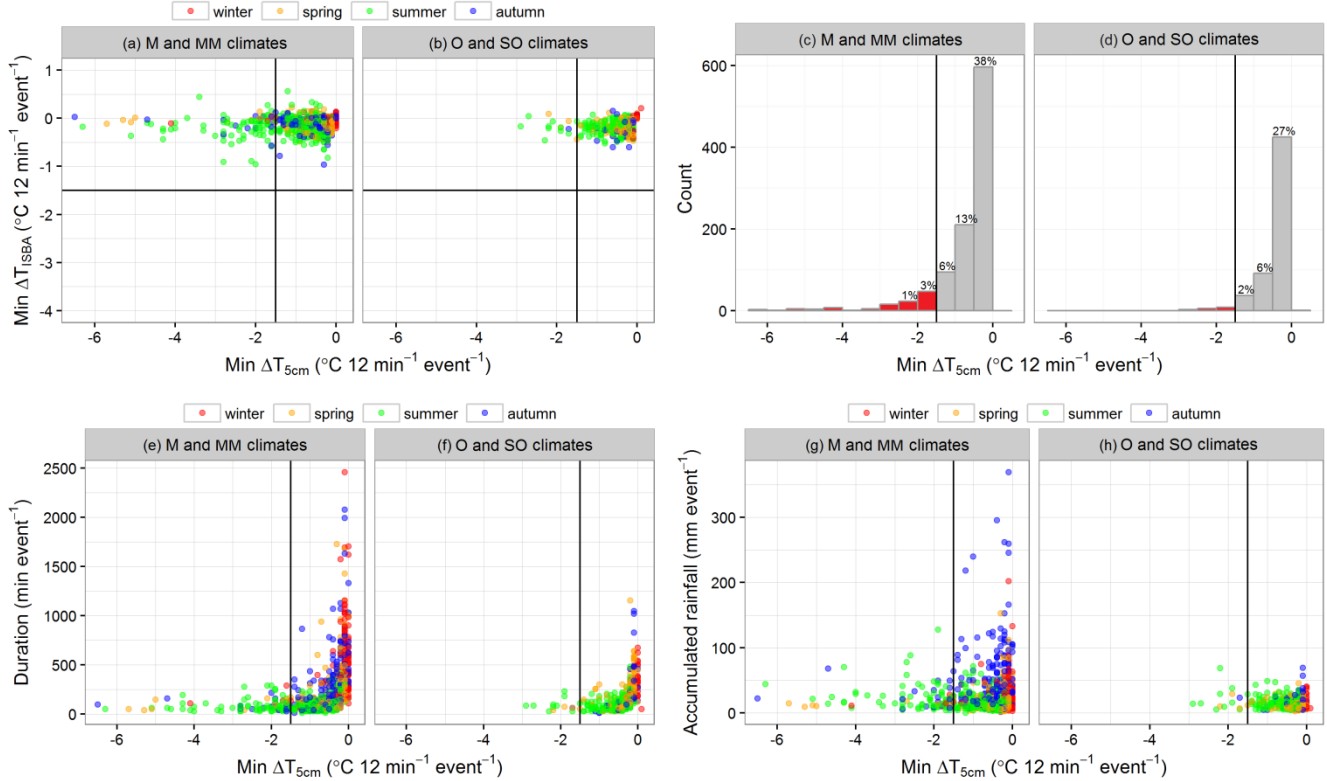

**Figure 6. Minimum $\Delta T_{5cm}$ during 1577 marked rainfall events affecting $T_{5cm}$: vs. minimum $\Delta T_{ISBA}$ (a, b), statistical distribution (bins of 0.5 ºC) (c, d), vs. rain duration (e, f), and vs. the accumulated rainfall (g, h), for Mediterranean (M) and Mediterranean-mountain (MM) stations (a, c, e, g) and for oceanic (O) and semi-oceanic (SO) stations (b, d, f, h). Dark lines are for the -1.5 °C threshold for intense soil-cooling rains (step 5 in Table 2).**





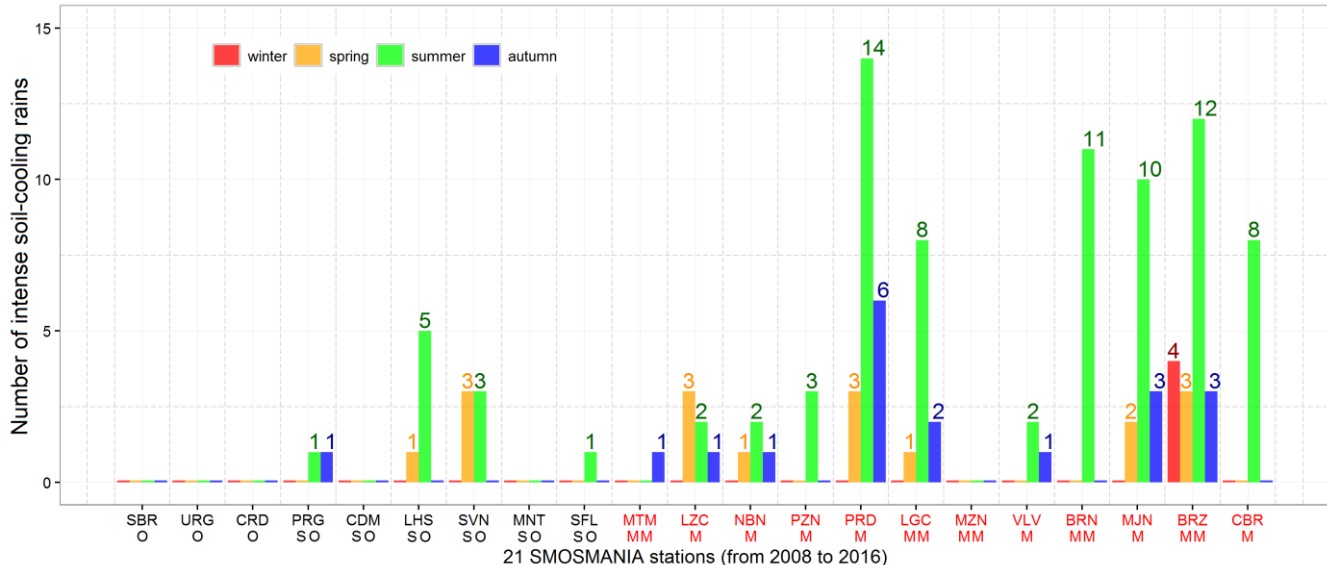

**Figure 7. Statistical distribution of the 122 intense soil-cooling rains among stations of the SMOSMANIA network across seasons. Red, orange, green and blue bars are for winter, spring, summer and the autumn, respectively. Symbols "O", "SO", "M", and "MM" indicate Oceanic, Semi-Oceanic, Mediterranean, and Mediterranean/Mountain climates, respectively. Stations are listed from (left) west to (right) east.**



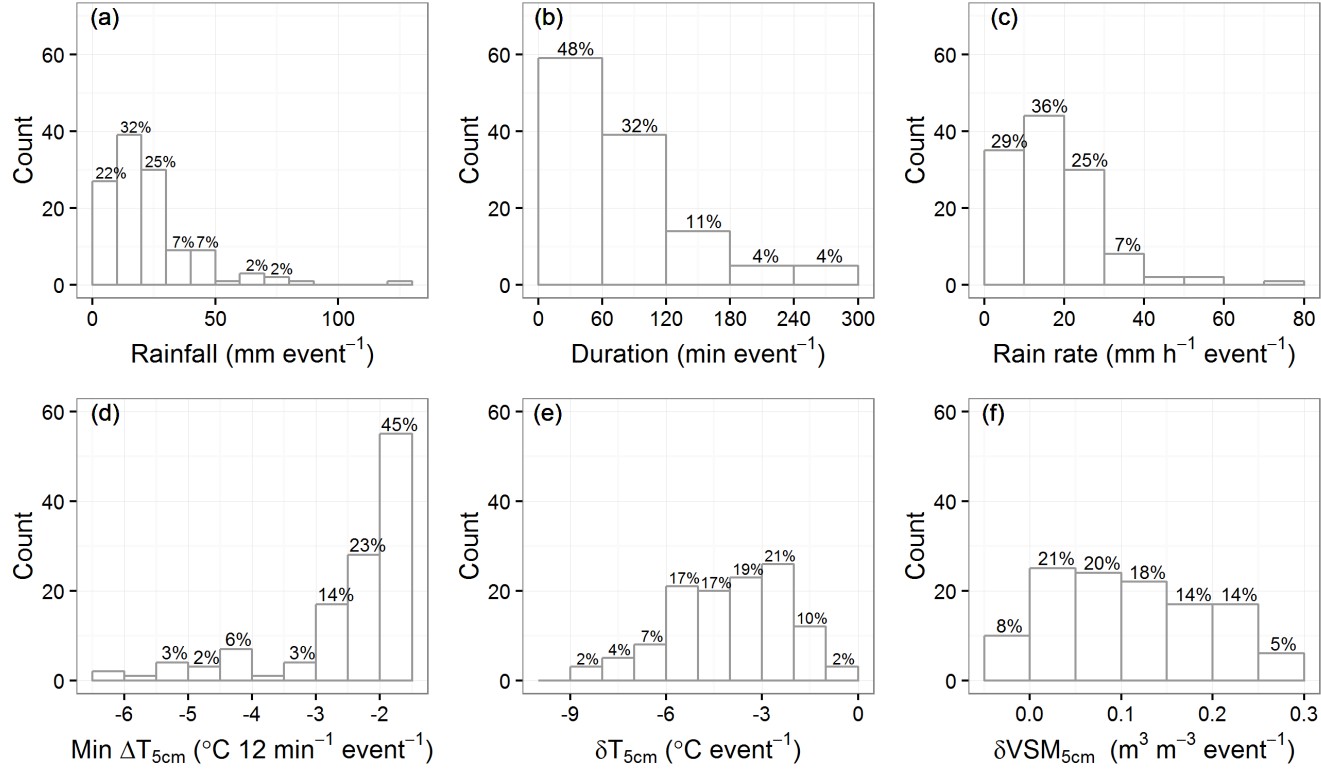

**Figure 8. Statistical distribution of the 122 intense soil-cooling rains in terms of (Sect. 3.3) accumulated rainfall (a), rain duration (b), rain rate (c), minimum $\Delta T_{5cm}$ (d), $\delta T_{5cm}$ (e), and $\delta VSM_{5cm}$ (f), with bins of 10 mm, 60 min, 10 mm h$^{-1}$, 1 °C, 1 °C, and 0.05 m$^3$ m$^{-3}$, respectively.**



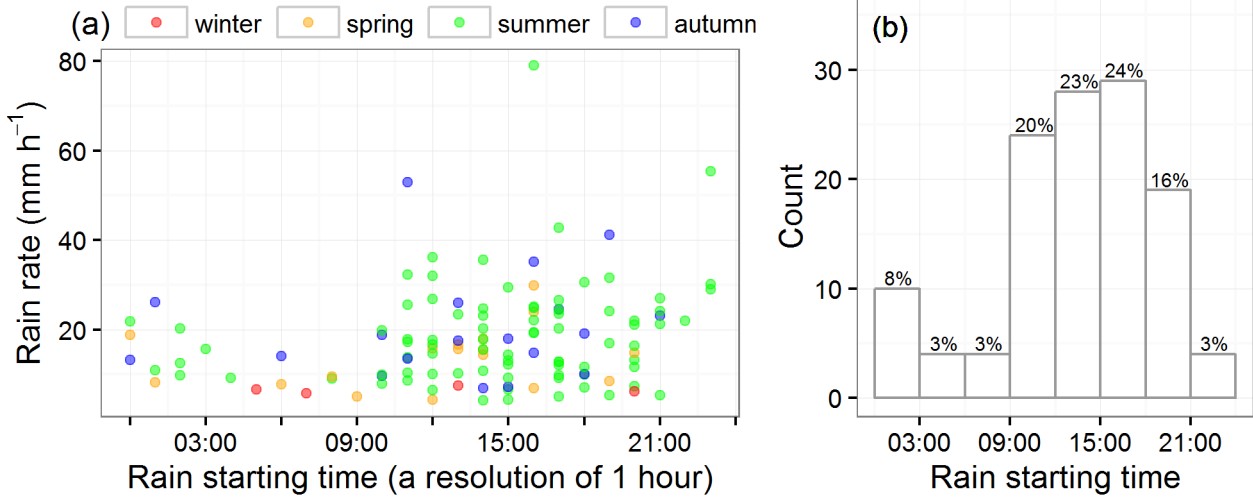

**Figure 9. Starting time of the 122 intense soil-cooling rains: vs. rain rate across seasons (a), statistical distribution (with bins of 3 hours).**





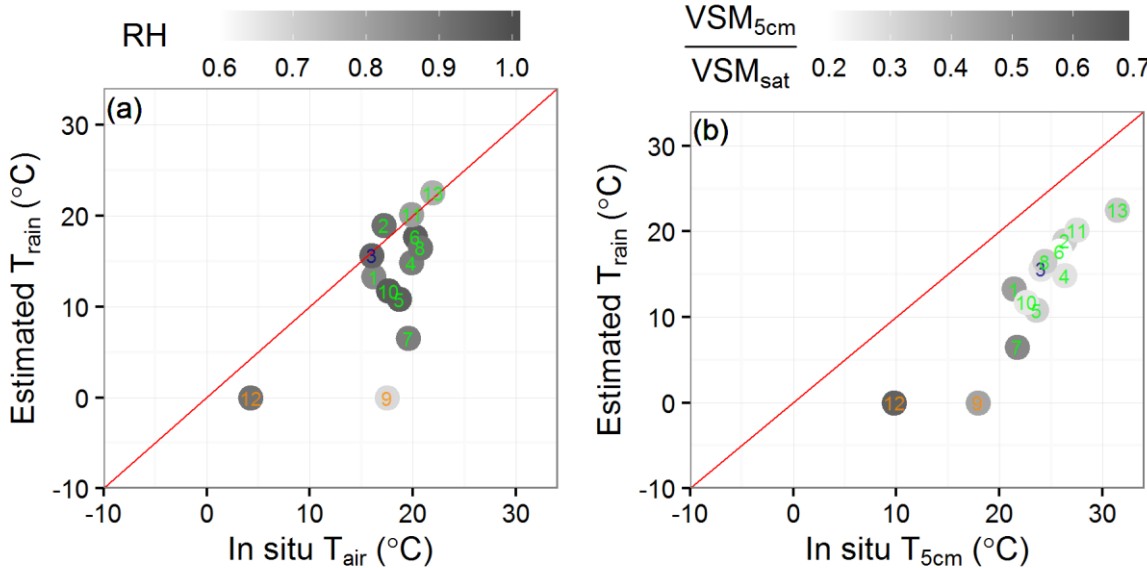

**Figure 10.** Estimated $T_{rain}$ for the 13 cases listed in Table 4 vs. observed ambient $T_{air}$ (a) and observed $T_{5cm}$ (b), with levels of grey indicating air relative humidity (RH, dimensionless), and $VSM_{5cm}$ to $VSM_{sat}$ ratio (dimensionless) values at time $t_1$, respectively.



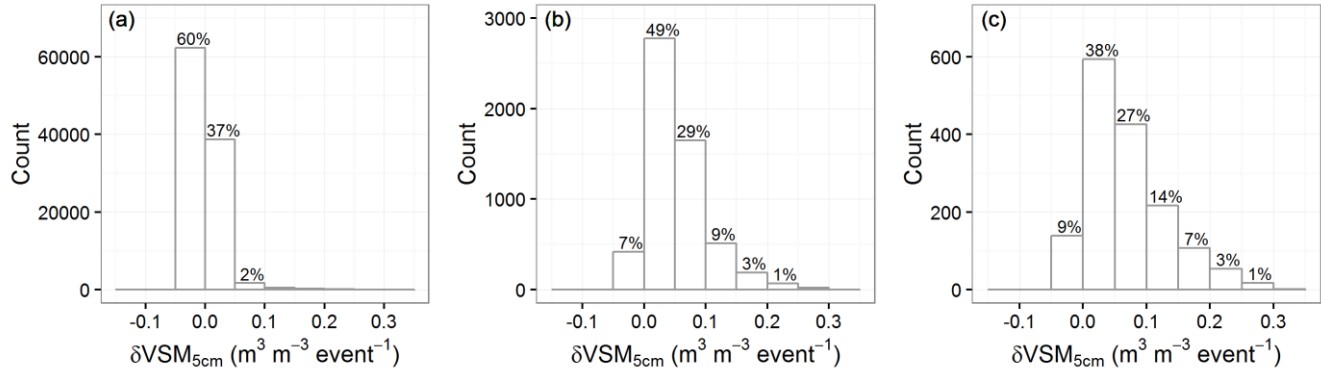

**Figure 11. Statistical distribution of $\delta VSM_{5cm}$ for: 104178 fully documented rainfall events (a), 5714 marked rainfall events (b), and 1577 marked rainfall events affecting topsoil temperature (c) defined in Table 2.**