# Peer review of "Identification of soil-cooling rains in southern France from soil temperature and soil moisture observations"

_Atmospheric Chemistry and Physics, 2018_

## Referee Comment (RC1) · Anonymous Referee #2 · 31 Dec 2018

General comments

The paper presents a study, based on in situ observations of soil temperature and soil moisture, that investigates the cooling effect of rainfall on soil water. In specific conditions, the paper also proposes a method to infer rainfall temperature from variations of soil temperature and soil moisture. The analysis relies on 9 years of 12-min time step data of soil temperature and soil water content (complemented by in situ climatic data from standard meteorological stations), collected in 21 stations of the southern part of France, encompassing various climatic conditions from oceanic to Mediterranean climates. The length of the time series is long enough to get sufficiently robust con-

clusions. As underlined by the authors themselves, data on rainfall temperature are rare – if not inexistent. The impact of potential differences between rainfall, air and soil temperature is generally not considered in atmospheric and climatic models, but it is worth trying to quantify if it is really negligible, or in which conditions it may be important to take into account this effect on the surface energy balance. Although the rainfall temperature estimation proposed in the paper is indirect and relies on hypotheses that are clearly stated by the authors, it can provide a first guess on rainfall temperature estimation to explore more in depth its impact on the surface energy balance. Besides its interest in documenting rainfall temperature, the paper also provides an interesting climatology of soil moisture and temperature changes when it rains. There is also an interesting approach for selecting and characterizing rainfall events, based on high temporal resolution data. Data presentation and analyses are clear and carefully performed, with a level of details that is suitable, although the presentation could sometimes be shortened (see details below). There is one point that is however not clear for me. The authors presents some simulations based on the ISBA land surface model, that does not include a representation of heat exchanges due to water mass movement, to show that it is not able to reproduce the observed change in soil temperature. To perform these simulations, the authors use the SAFRAN reanalysis that has a low temporal resolution for rainfall (hourly) and a spatial resolution of 8x8 km2. This resolution may not allow the forcing to capture very localized rainfall events. If in situ meteorological data are available, why didn't the authors use them?

Specific comments 1/ p.4, lines 10-14; p.8 lines 10-16: why do you use SAFRAN reanalysis and not the in situ collected meteorological data? The low resolution of SAFRAN is probably not adapted to capture the high spatial and temporal variability of rainfall that is relevant if you want to relate the forcing to temperature and soil moisture local variations. 2/ Section 3.3: the location of this section is strange. Why didn't you put it in an appendix? 3/ Section 4.2: this section is very detailed. It could be shortened. Fig. 9 could also be put in the supplementary materials. 4/ Section 5.2. This section is somehow frustrating: the authors have gathered all the data to test to impact of

cold rainfall temperature on the soil temperature and water content and the reader is expecting to see such a simulation using the ISBA model. Adding a representation of heat exchanges due to water mass movement into the model could be done in order to complement the paper. It would also give more strength to the conclusions on whether rainfall cooling matters or not.
* * *

---

## Referee Comment (RC2) · Anonymous Referee #3 · 27 Jan 2019

General comment: This paper presents an assessment of soil-cooling rain events in South of France and is based on observations recorded during 9 years, a long enough period to allow robust statistics. The paper is mostly a description of the dataset which is stratified in different ways. The dataset and the method are generally well described and the argument is quite relevant. The modelling aspects are less satisfactory, for instance, the comparison between ISBA and the observations is not convincing since ISBA does not represent the cooling process and the quality of the forcing is poor ( duration and intensity of the rain events). On one hand the discussion of the results could have been shortened may be summarizing some of them in tables, on the other hand insights to understand when, where and why soil cooling occurs or not would

have been valuable to help model development. For instance section 5.2 starts well "Does soil cooling matter" but at the end of the section it is not clear what is the added value of the paper to answer this question.

Minor comments The meaning of the sentence starting line 29 in section 3.1 is not clear.

Section 3.2 l. 24 : Why the precipitation induced sensible heat flux dominates the heat exchange, it it possible to evaluate it and compare with the heat conduction?

Section 4.1 When speaking about the minimum deltaT5cm using absolute values may render easier the reading:: even if it's correct: "larger than -0.5C" is a bit confusing.

Panels in Figure 6 are partially commented, if they are not essential they have to be removed or put in the supplemetary materials. By the way, units are original!

---

## Author Comment (AC1) · 26 Feb 2019

RESPONSE TO REVIEWER #2

The authors thank anonymous reviewer 2 for his/her review of the manuscript and for the fruitful comments.

2.1 [General comments The paper presents a study, based on in situ observations of soil temperature and soil moisture, that investigates the cooling effect of rainfall on soil water. In specific conditions, the paper also proposes a method to infer rainfall temperature from variations of soil temperature and soil moisture. The analysis relies

on 9 years of 12-min time step data of soil temperature and soil water content (complemented by in situ climatic data from standard meteorological stations), collected in 21 stations of the southern part of France, encompassing various climatic conditions from oceanic to Mediterranean climates. The length of the time series is long enough to get sufficiently robust conclusions. As underlined by the authors themselves, data on rainfall temperature are rare – if not inexistent. The impact of potential differences between rainfall, air and soil temperature is generally not considered in atmospheric and climatic models, but it is worth trying to quantify if it is really negligible, or in which conditions it may be important to take into account this effect on the surface energy balance. Although the rainfall temperature estimation proposed in the paper is indirect and relies on hypotheses that are clearly stated by the authors, it can provide a first guess on rainfall temperature estimation to explore more in depth its impact on the surface energy balance. Besides its interest in documenting rainfall temperature, the paper also provides an interesting climatology of soil moisture and temperature changes when it rains. There is also an interesting approach for selecting and characterizing rainfall events, based on high temporal resolution data. Data presentation and analyses are clear and carefully performed, with a level of details that is suitable, although the presentation could sometimes be shortened (see details below).]

Response 2.1:

Many thanks for this positive evaluation of our work.

2.2 [General comments There is one point that is however not clear for me. The authors presents some simulations based on the ISBA land surface model, that does not include a representation of heat exchanges due to water mass movement, to show that it is not able to reproduce the observed change in soil temperature. To perform these simulations, the authors use the SAFRAN reanalysis that has a low temporal resolution for rainfall (hourly) and a spatial resolution of 8x8 km2. This resolution may not allow the forcing to capture very localized rainfall events. If in situ meteorological data are available, why didn't the authors use them? Specific comments 1/ p.4, lines 10-14; p.8

lines 10-16: why do you use SAFRAN reanalysis and not the in situ collected meteorological data? The low resolution of SAFRAN is probably not adapted to capture the high spatial and temporal variability of rainfall that is relevant if you want to relate the forcing to temperature and soil moisture local variations.]

Response 2.2:

The shown ISBA simulations represent the current state of hourly operational land surface monitoring, available over whole of metropolitan France. The message we want to convey is that the best possible operational simulations currently available are not able to represent the impact of intense precipitation on the soil temperature profile. The ISBA land surface model needs to be improved. The SAFRAN atmospheric analysis could also probably be improved by using more in situ observations together with high resolution atmospheric simulations. This is work in progress. Unfortunately, local meteorological data do not include all the atmospheric variables needed to force the ISBA land surface model. Using locally observed precipitation together with other variables from SAFRAN would not be technically correct. Therefore, properly disentangling SAFRAN and ISBA shortcomings is not possible now. We understand that this can be confusing for the readers: the ISBA simulations will be moved to the Supplement, including Figure 6.

2.3 [ 2/ Section 3.3: the location of this section is strange. Why didn't you put it in an appendix? ]

Response 2.3:

Yes. We will move Section 3.3 to an appendix.

2.4 [3/ Section 4.2: this section is very detailed. It could be shortened. Fig. 9 could also be put in the supplementary materials. ]

Response 2.4:

Yes. We will shorten Section 4.2, and move Fig. 9 to the supplement. The content of

Fig. 9 can easily be summarized in a small Table.

2.5 [4/ Section 5.2. This section is somehow frustrating: the authors have gathered all the data to test the impact of cold rainfall temperature on the soil temperature and water content and the reader is expecting to see such a simulation using the ISBA model. Adding a representation of heat exchanges due to water mass movement into the model could be done in order to complement the paper. It would also give more strength to the conclusions on whether rainfall cooling matters or not. ]

Response 2.5:

Section 5.2 shows that rain water temperature is needed for a number of applications. Now, the ISBA model has no representation of heat exchanges due to water mass movement. This process needs to be introduced in ISBA. We think that data from a fully instrumented site including direct measurements of rain water temperature are needed to completely address this issue and to validate the upgraded model version. Such an experiment would give insights to understand when, where and why soil cooling occurs or not and would be valuable to help model development. In particular, the precipitation-induced sensible heat flux is not limited to intense precipitation and the impact of this process on the surface energy budget needs to be investigated in all conditions. We plan to perform these tasks in future works.

See also Response 2.2. This discussion will be included in Section 5.2.
* * *

---

## Author Comment (AC2) · 26 Feb 2019

RESPONSE TO REVIEWER #3

The authors thank anonymous reviewer 3 for his/her review of the manuscript and for the fruitful comments.

3.1 [General comment: This paper presents an assessment of soil-cooling rain events in South of France and is based on observations recorded during 9 years, a long enough period to allow robust statistics. The paper is mostly a description of the dataset which is stratified in different ways. The dataset and the method are gener-

ally well described and the argument is quite relevant. The modelling aspects are less satisfactory, for instance, the comparison between ISBA and the observations is not convincing since ISBA does not represent the cooling process and the quality of the forcing is poor (duration and intensity of the rain events). On one hand the discussion of the results could have been shortened may be summarizing some of them in tables, on the other hand insights to understand when, where and why soil cooling occurs or not would have been valuable to help model development. For instance section 5.2 starts well "Does soil cooling matter" but at the end of the section it is not clear what is the added value of the paper to answer this question.]

Response 3.1:

See also Responses 2.2 and 2.5.

Now, the ISBA model has no representation of heat exchanges due to water mass movement. This process needs to be introduced in ISBA. The shown ISBA simulations represent the current state of hourly operational land surface monitoring, available over whole of metropolitan France. The message we want to convey is that the best possible operational simulations currently available are not able to represent the impact of intense precipitation on the soil temperature profile. The ISBA land surface model needs to be improved. The SAFRAN atmospheric analysis could also probably be improved by using more in situ observations together with high resolution atmospheric simulations. This is work in progress. Unfortunately, local meteorological data do not include all the atmospheric variables needed to force the ISBA land surface model. Using locally observed precipitation together with other variables from SAFRAN would not be technically correct. Therefore, properly disentangling SAFRAN and ISBA shortcomings is not possible now. We understand that this can be confusing for the readers: the ISBA simulations will be moved to the Supplement, including Figure 6. Section 5.2 shows that rain water temperature is needed for a number of applications. We think that data from a fully instrumented site including direct measurements of rain water temperature are needed to completely address this issue and to validate the upgraded

model version. Such an experiment would give insights to understand when, where and why soil cooling occurs or not and would be valuable to help model development. In particular, the precipitation-induced sensible heat flux is not limited to intense precipitation and the impact of this process on the surface energy budget needs to be investigated in all conditions. We plan to perform these tasks in future works.

This discussion will be included in Section 5.2.

3.2 [Minor comments The meaning of the sentence starting line 29 in section 3.1 is not clear.]

Response 3.2:

"The missing data fraction across seasons (Table S1) is used to correct the estimation of the possible number of intense soil-cooling rainfall events, their frequency and the mean time lag between two events."

This sentence refers to Table 1. It will be rephrased and moved to Section 2, where Table 1 is presented.

3.3 [Section 3.2 l. 24 : Why the precipitation induced sensible heat flux dominates the heat exchange, is it possible to evaluate it and compare with the heat conduction?]

Response 3.3:

Since soil properties are known, the mean precipitation-induced sensible heat flux $P_h$ can be estimated from Eq. (2) for the intense soil-cooling events used to retrieve $T_{rain}$ (see Table 4). For the 10 events of Table 4 occurring at summertime, this flux ranges from 408 to 1009 W m-2, with a mean value of 648 W m-2. These $P_h$ flux values are very high and represent large fractions of the net radiation $R_{net}$ (i.e. the amount of energy available for surface heat exchanges, driven by the incoming solar radiation, that could be simulated without accounting for $P_h$). They are probably often much larger than $R_{net}$ because the $R_{net}$ energy budget component is generally small during rainfall events, in relation to the low incoming solar radiation. Moreover, 7 events out

of 10 occur at nighttime or at dusk (see Supplement), i.e. in small R_net value conditions. R_net is not measured at SMOSMANIA stations. Typical measured summertime values of the maximum daily R_net over the grassland site of Meteopole-Flux (Zhang et al. 2018) in southwestern France range from about 200 W m-2 during cloudy rainy days to about 700 W m-2 in clear sky conditions. Minimum R_net values at nighttime range from -100 to 0 W m-2.

This discussion will be included in Section 5.1.

3.4 [Section 4.1 When speaking about the minimum deltaT5cm using absolute values may render easier the reading:: even if it's correct: "larger than -0.5C" is a bit confusing.]

Response 3.4:

Yes, "are larger than -0.5 $^\circ$C in 12 minutes" will be replaced by "do not depart much from 0 $^\circ$C in 12 minutes".

3.5 [Panels in Figure 6 are partially commented, if they are not essential they have to be removed or put in the supplemetary materials. By the way, units are original!]

Response 3.5:

Yes, we will move Figure 6 to the supplement.

---

## Author Response (AR1)

**Zhang et al.: Identification of soil-cooling rains in southern France from soil temperature and soil moisture observations, Atmos. Chem. Phys. Discuss., https://doi.org/10.5194/acp-2018-929.**

**RESPONSE TO REVIEWER #2**

2.2 [General comments
There is one point that is however not clear for me. The authors presents some simulations based on the ISBA land surface model, that does not include a representation of heat exchanges due to water mass movement, to show that it is not able to reproduce the observed change in soil temperature. To perform these simulations, the authors use the SAFRAN reanalysis that has a low temporal resolution for rainfall (hourly) and a spatial resolution of 8x8 km2. This resolution may not allow the forcing to capture very localized rainfall events. If in situ meteorological data are available, why didn't the authors use them?
Specific comments
1/ p.4, lines 10-14; p.8 lines 10-16: why do you use SAFRAN reanalysis and not the in situ collected meteorological data? The low resolution of SAFRAN is probably not adapted to capture the high spatial and temporal variability of rainfall that is relevant if you want to relate the forcing to temperature and soil moisture local variations.]

**Response 2.2:**

The shown ISBA simulations represent the current state of hourly operational land surface monitoring, available over whole of metropolitan France. The message we want to convey is that the best possible operational simulations currently available are not able to represent the impact of intense precipitation on the soil temperature profile. The ISBA land surface model needs to be improved. The SAFRAN atmospheric analysis could also probably be improved by using more in situ observations together with high resolution atmospheric simulations. This is work in progress. Unfortunately, local meteorological data do not include all the atmospheric variables needed to force the ISBA land surface model. Using locally observed precipitation together with other variables from SAFRAN would not be technically correct. Therefore, properly disentangling SAFRAN and ISBA shortcomings is not possible now. We understand that this can be confusing for the readers: **the ISBA simulations were moved to the Supplement, including Figure 6.**

2.3 [ 2/ Section 3.3: the location of this section is strange. Why didn't you put it in an appendix? ]

**Response 2.3:**

**Section 3.3 was moved to an appendix.**

2.4 [3/ Section 4.2: this section is very detailed. It could be shortened. Fig. 9 could also be put in the supplementary materials. ]

**Response 2.4:**

**Section 4.2 was shortened: last paragraph of this section and Fig. 9 were moded to the supplement. The content of old Fig. 9 was summarized in new Table 4.**

**Table 4.** Characteristics of the 122 intense soil-cooling rains.

| Characteristic | Event count or fraction |
|---|---|
| M or MM climate | 107 |
| O or SO climate | 15 |
| Spring | 17 |
| Summer | 82 |
| Autumn | 19 |
| Winter | 4 |
| Starting time at daytime (9:00 am – 21:00 pm) | 83 % |
| Duration less than 2 hours | 80 % |
| Minimum $\Delta T_{5cm}$ values ranging from -3 to -1.5 ºC 12 min$^{-1}$ | 82 % |
| Maximum number of events per year per station | 2.7 |

2.5 [4/ Section 5.2. This section is somehow frustrating: the authors have gathered all the data to test the impact of cold rainfall temperature on the soil temperature and water content and the reader is expecting to see such a simulation using the ISBA model. Adding a representation of heat exchanges due to water mass movement into the model could be done in order to complement the paper. It would also give more strength to the conclusions on whether rainfall cooling matters or not. ]

**Response 2.5:**

**The following discussion was included in Section 5.2:**
"Now, the ISBA model has no representation of heat exchanges due to water mass movement. This process needs to be introduced in ISBA. We think that data from a fully instrumented site including direct measurements of rain water temperature are needed to completely address this issue and to validate the upgraded model version. Such an experiment would give insights to understand when, where and why soil cooling occurs or not and would be valuable to help model development. In particular, the precipitation-induced sensible heat flux is not limited to intense precipitation and the impact of this process on the surface energy budget needs to be investigated in all conditions."

**RESPONSE TO REVIEWER #3**

3.1 [General comment: This paper presents an assessment of soil-cooling rain events in South of France and is based on observations recorded during 9 years, a long enough period to allow robust statistics. The paper is mostly a description of the dataset which is stratified in different ways. The dataset and the method are generally well described and the argument is quite relevant. The modelling aspects are less satisfactory, for instance, the comparison between ISBA and the observations is not convincing since ISBA does not represent the cooling process and the quality of the forcing is poor (duration and intensity of the rain events). On one hand the discussion of the results could have been shortened may be summarizing some of them in tables, on the other hand insights to understand when, where and why soil cooling occurs or not would have been valuable to help model development. For instance section 5.2 starts well "Does soil cooling matter" but at the end of the section it is not clear what is the added value of the paper to answer this question.]

**Response 3.1:**

See also Responses 2.2 and 2.5.

**The ISBA simulations were moved to the Supplement, including Figure 6.**

**The following discussion was included in Section 5.2:**
"Now, the ISBA model has no representation of heat exchanges due to water mass movement. This process needs to be introduced in ISBA. We think that data from a fully instrumented site including direct measurements of rain water temperature are needed to completely address this issue and to validate the upgraded model version. Such an experiment would give insights to understand when, where and why soil cooling occurs or not and would be valuable to help model development. In particular, the precipitation-induced sensible heat flux is not limited to intense precipitation and the impact of this process on the surface energy budget needs to be investigated in all conditions."

3.2 [Minor comments The meaning of the sentence starting line 29 in section 3.1 is not clear.]

**Response 3.2:**

"The missing data fraction across seasons (Table S1) is used to correct the estimation of the possible number of intense soil-cooling rainfall events, their frequency and the mean time lag between two events."

This sentence refers to Table 1. **It was rephrased and moved to Section 2, where Table 1 is presented:**

"Table 1 also presents the frequency of intense soil-cooling events (see Sections 3 and 4). Since a noticeable fraction of observed T5cm or VSM5cm data is missing, the number of marked soil-cooling rainfall events could be underestimated. The missing data fraction across

seasons (Table S1) is used to correct the estimation of the possible number of intense soil-cooling rainfall events, their frequency and the mean time lag between two events in Table 1."

3.3 [Section 3.2 l. 24 : Why the precipitation induced sensible heat flux dominates the heat exchange, is it possible to evaluate it and compare with the heat conduction?]

**Response 3.3:**

**The following discussion was included in Section 5.1:**

"In Eqs. (2) and (3), it is assumed that the precipitation-induced sensible heat flux dominates heat exchanges in the topsoil layer. Since soil properties are known, the mean PH value can be estimated from Eq. (2) for the intense soil-cooling events used to retrieve Train (see Table 5). For the 10 events of Table 5 occurring at summertime, this flux ranges from 408 to 1009 W m-2, with a mean value of 648 W m-2. These PH flux values are very high and represent large fractions of absolute values of the net radiation Rnet (i.e. the amount of energy available for surface heat exchanges, driven by the incoming solar radiation, that could be simulated without accounting for PH). They are probably often much larger than Rnet because the Rnet energy budget component is generally small during rainfall events, in relation to the low incoming solar radiation. Moreover, 7 events out of 10 occur at nighttime or at dusk (see Supplement), i.e. in small Rnet value conditions. The Rnet variable is not measured at SMOSMANIA stations. Typical measured summertime values of the maximum daily Rnet over the grassland site of Meteopole-Flux in southwestern France (Zhang et al. 2018) range from about 200 W m-2 during cloudy rainy days to 5 about 700 W m-2 in clear sky conditions. At nighttime, absolute Rnet values rarely exceed 100 W m-2."

3.4 [Section 4.1 When speaking about the minimum deltaT5cm using absolute values may render easier the reading:: even if it's correct: "larger than -0.5C" is a bit confusing.]

**Response 3.4:**

**"are larger than -0.5 °C in 12 minutes" was replaced by "do not depart much from 0 °C in 12 minutes" (now in Supplement P. 28).**

3.5 [Panels in Figure 6 are partially commented, if they are not essential they have to be removed or put in the supplemetary materials. By the way, units are original!]

**Response 3.5:**

**Figure 6 was moved to the Supplement (now Figure S25).**

[revised manuscript text omitted]
 shown ISBA simulations represent the current state of hourly operational land surface monitoring, available over whole of metropolitan France. This means that the best possible operational simulations currently available are not able to represent the impact of intense precipitation on the soil temperature profile. The ISBA land surface model needs to be improved. The SAFRAN atmospheric analysis could also probably be improved by using more in situ observations together with high resolution atmospheric simulations.

[Figure]

**Figure S24.** ISBA simulations of soil temperature (left) and soil moisture (right) at the PRD station from 21 to 25 August 2015 at depths of 5, 10, 20 and 30 cm, together with the SAFRAN rainfall data (mm hour-1) shown in grey.

Figure S25 presents the statistical distribution of minimum $\Delta T_{5cm}$ observations, and the corresponding minimum $\Delta T_{5cm}$ values simulated by the ISBA model for the 1577 marked rainfall events affecting $T_{5cm}$. It appears that step 5 tends to remove the longest rainfall events and the selected rainfall events last less than 4 hours. The comparison between observed and simulated values shows that ISBA is not able to simulate $\Delta T_{5cm}$ values well. In particular, most of the simulated minimum $\Delta T_{5cm}$ values do not depart much from 0 °C in 12 minutes during intense soil-cooling events, even for very intense ones with observed minimum $\Delta T_{5cm}$ values lower than -4 °C in 12 minutes. Figure S25 also shows that most of the 122 intense soil-cooling events occur in M or in MM climate conditions.

[Figure]

**Figure S25**. Minimum $\Delta T_{5cm}$ during 1577 marked rainfall events affecting $T_{5cm}$: vs. minimum $\Delta T_{ISBA}$ (a, b), statistical distribution (bins of 0.5 ℃) (c, d), vs. rain duration (e, f), and vs. the accumulated rainfall (g, h), for Mediterranean (M) and Mediterranean-mountain (MM) stations (a, c, e, g) and for oceanic (O) and semi-oceanic (SO) stations (b, d, f, h). Dark lines are for the -1.5 °C threshold for intense soil-cooling rains (step 5 in Table 2).